# Myr-Arf1 conformational flexibility at the membrane surface sheds light on the interactions with ArfGAP ASAP1

Yue Zhang [1,10,13], Olivier Soubias[1,13], Shashank Pant [2,11,13], Frank Heinrich [3,4], Alexander Vogel [5], Jess Li[1], Yifei Li[1,12], Luke A. Clifton [6], Sebastian Daum[7], Kirsten Bacia[7], Daniel Huster [5], Paul A. Randazzo [8], Mathias Lösche[3,4,9], Emad Tajkhorshid[2] & R. Andrew Byrd [1] ✉

ADP-ribosylation factor 1 (Arf1) interacts with multiple cellular partners and membranes to regulate intracellular traffic, organelle structure and actin dynamics. Defining the dynamic conformational landscape of Arf1 in its active form, when bound to the membrane, is of high functional relevance and key to understanding how Arf1 can alter diverse cellular processes. Through concerted application of nuclear magnetic resonance (NMR), neutron reflectometry (NR) and molecular dynamics (MD) simulations, we show that, while Arf1 is anchored to the membrane through its N-terminal myristoylated amphipathic helix, the G domain explores a large conformational space, existing in a dynamic equilibrium between membrane-associated and membrane-distal conformations. These configurational dynamics expose different interfaces for interaction with effectors. Interaction with the Pleckstrin homology domain of ASAP1, an Arf-GTPase activating protein (ArfGAP), restricts motions of the G domain to lock it in what seems to be a conformation exposing functionally relevant regions.

Small GTPases of the RAS-superfamily associate with membranes through lipid recognition and insertion of lipid anchors where they form transient complexes that drive signaling important for normal physiology. In turn, functional aberration of these GTPases can result in disease, including cancer. Among them, ADP-ribosylation factor-1 (Arf1) is a small, lipidated GTPase that regulates membrane traffic, organelle structure, and actin dynamics by cycling between a cytosolic,

GDP-bound state and a membrane-associated, GTP-bound state. Structural studies have shown that Arf1 comprises a canonical GTPase domain (G domain), which includes the nucleotide-binding site and predicted effector-binding regions (EBRs), switch I, switch II, and α−3 helix, connected by a short unstructured linker to a myristoylated N-terminal amphipathic α-helix[1]. N-myristoylation (myr) and the N-terminal α-helix of Arf1 are essential for biological function. In

[1]Center for Structural Biology, Center for Cancer Research, National Cancer Institute, Frederick, MD 21702-1201, USA. [2]Theoretical and Computational Biophysics Group, NIH Center for Macromolecular Modeling and Visualization, Beckman Institute for Advanced Science and Technology, Department of Biochemistry, and Center for Biophysics and Quantitative Biology, University of Illinois at Urbana-Champaign, Urbana, IL 61801, USA. [3]Department of Physics, Carnegie Mellon University, Pittsburgh, PA, USA. [4]NIST Center for Neutron Research, Gaithersburg, MD, USA. [5]Institute of Medical Physics and Biophysics, University of Leipzig, 04107 Leipzig, Germany. [6]ISIS Neutron and Muon Source, Rutherford Appleton Laboratory, Didcot, Oxfordshire OX11 0QX, UK. [7]Institute for Chemistry, Department of Biophysical Chemistry, Martin Luther University Halle-Wittenberg, Kurt-Mothes-Str. 3A, 06120 Halle, Germany. [8]Laboratory of Cellular and Molecular Biology, National Cancer Institute, National Institutes of Health, Bethesda, MD 20892, USA. [9]Department of Biomedical Engineering, Carnegie Mellon University, Pittsburgh, PA, USA. [10]Present address: Ring Therapeutics, Inc., Cambridge, MA, USA. [11]Present address: Loxo Oncology at Lilly, Louisville, CO, USA. [12]Present address: Vonsun Pharmatech Co., Ltd., Suzhou, China. [13]These authors contributed equally: Yue Zhang, Olivier Soubias, Shashank Pant. ✉e-mail: byrdra@nih.gov

myr-Arf1•GDP, the myristate occupies a hydrophobic cleft in the protein, formed by switch I (sw1), switch II (sw2), and the interswitch regions, and is shielded by disordered N-terminal residues[2]. Exchange of GTP for GDP results in conformational changes in sw1, sw2, and interswitch regions, with a reduction in the surface providing the hydrophobic cleft. Myristate is consequently ejected from the protein, and, together with the N-terminal residues that form an α-helix, associates with a membrane surface[3]. The association is necessary for stability of myr-Arf1•GTP, which will not form to a measurable extent without a membrane[4–6].

Nuclear magnetic resonance (NMR) studies of yeast myr-Arf1 in the presence of lipid bicelles indicated that the G domain was in dynamic equilibrium between different conformations[3]. The same study confirmed earlier NMR and neutron studies of the isolated myristoylated peptide[7–10] showing that the N-terminal helix lies with its axis approximately parallel to the membrane surface. However, the influence of lipids on the conformational space explored by the G domain could not be examined and the myristoylated acyl chain was found to be folded back against the N-term helix, parallel to the membrane surface. The wide range of effectors recruited to the membrane surface by Arf[11] suggests that Arf1 may need to present different interfaces to orchestrate cell signaling in a spatially and temporally regulated manner.

In this study, we combine NMR and neutron reflectometry (NR) with molecular dynamics (MD) simulations to investigate the interaction and dynamics of human myr-Arf1•GTP with model lipid membranes of relevant compositions. Our results show how myr-Arf1•GTP binds the membrane surface through its myristoylated N-terminal helix. The myristoyl acyl chain inserts into the lipid matrix, and the amphipathic N-terminal helix buries in the membrane surface. The G domain remains highly dynamic but predominantly explores three differentially populated states in the presence of anionic lipids, each exposing different interfaces for potential interaction with effector proteins but with a distribution only marginally dependent on

membrane surface charge. Interestingly, binding the pleckstrin homology (PH) domain of the GTPase activating protein (GAP) ArfGAP with SH3 domain, ankyrin repeat, and PH domain 1 (ASAP1)[12–15] drastically alters the dynamic equilibrium of the G domain, suggesting a stepwise assembly of the multivalent complex between Arf and ASAP1 at the membrane surface, wherein PH coincident recognition of a PI(4,5)P$_2$ headgroup[16,17] and Arf1 are the first steps toward GTP hydrolysis.

## Results

### High yield expression of isotopically labeled myr-Arf1

NMR studies of the structure and dynamics of membrane-bound Arf1 under functional conditions require the expression of an isotopically labeled, perdeuterated, and N-myristoylated protein. Efficient N-myristoylation of Arf1 (Fig. 1A and Supplementary Fig. 1) combined with U-$^{15}$N and $^{13}$C-methyl labeling (myr-Arf1) was reported previously[18]; however, expression was very inefficient when combined with normal perdeuteration procedures[19]. Conversion of the prior dual vector system to a single pETDuet-1 (Novagen) system[20] combined with perdeuteration[19] led to a ten-fold improvement (7–8 mg per liter of culture) in production of U-$^2$H and $^{13}$C-methyl labeled myr-Arf1 (see "Methods" and Supplementary Fig. 3). In the reported NMR studies, isotopically labeled myr-Arf1 is reconstituted in membrane-mimetic nanodiscs (ND) with the scaffold protein MSPΔH5[18] (Supplementary Figs. 2, 3), a well-established membrane mimetic to facilitate solution-state NMR spectroscopy of integral membrane proteins and membrane surface complexes[21–23].

### Arf1 G domain remains dynamic at the membrane

Both $^{15}$N–transverse relaxation optimized spectroscopy (TROSY) and heteronuclear single quantum coherence (HSQC) spectra of a nanodisc-bound U-[$^2$H,$^{15}$N] myr-Arf1•GTP, myr-Arf1•GTP:nanodisc, at 25 °C showed remarkably sharp and well-resolved peaks for a 130-kDa lipid/protein complex, which is surprising (Fig. 1B, D). All $^1$H-$^{15}$N cross peaks expected for the Arf1 G domain[18] were observed; however, resonances stemming from the N-terminal helix (residues 2–15) were missing, indicating distinct rotational correlation times for the G domain (~21 kDa, mobile) and the N-terminal helix (bound to nanodisc, 90 kDa). The mobility of the G domain is illustrated by comparing the rotational correlation time, $\tau_c$, measured by 1D TROSY for rotational correlation time (TRACT) experiments to the calculated $\tau_c$ based on the molecular weight of the complex. The $\tau_c$ was experimentally determined to be 34 ns for the G domain of Arf1•GTP bound to nanodisc. A protein the size of Arf1, free in solution, should exhibit a $\tau_c$ ~ 13 ns[24]. If Arf1 was bound tightly to the nanodisc, it would be presumed to take on $\tau_c$ of the ~130-kDa complex of ~60 ns[25]. For comparison, an empty MSPΔH5 nanodisc was reported to have a $\tau_c$ of 34 ns at 45 °C[26], which correlates to ~55 ns at 25 °C[27]. On the contrary, $^1$H-$^{13}$C methyl Heteronuclear Multiple Quantum Coherence (HMQC) spectrum of δ1-$^{13}$C$^1$H$_3$-labeled isoleucine (I), δ1-$^{13}$C$^1$H$_3$-labeled Leucine (L) and γ1-$^{13}$C$^1$H$_3$-labeled Valine (V) and otherwise perdeuterated myr-Arf1•GTP:nanodisc showed high-quality spectra (Fig. 1C) for the residues in the G domain and the N-terminal helix (I4, L8, and L11), confirming the binding of the N-terminal helix to the nanodisc.

### NR reveals multiple distal orientations of the G domain

Inherently a low-resolution surface-sensitive technique, NR yields one-dimensional spatial distributions, expressed as component volume occupancy (CVO) profiles, of substrate-supported lipid membranes and membrane-associated protein along the membrane normal[23,28]. CVO profiles thus quantify the volume within a plane at a certain distance, z, from the interface, occupied by lipid and protein components (cf ref. 16). Here, NR experiments were conducted with 85:15 POPC:POPS, 95:5 POPC:PI(4,5)P$_2$, and 87:10:3 POPC:POPS:PI(4,5)P$_2$ (see "Methods" for lipid abbreviations) sparsely tethered bilayer lipid

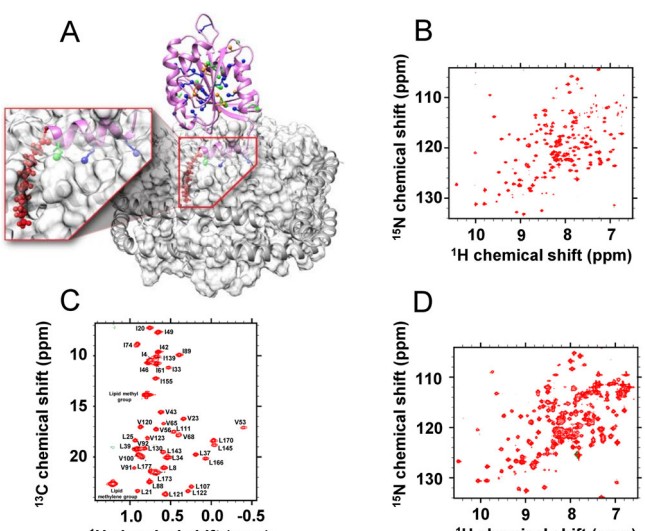

**Fig. 1 | myr-Arf1 bound to nanodisc and characterized by NMR. A** Homology model of human myr-Arf1, generated based on yeast Arf1 (PDB:2KSQ) using MODELLER, is shown in magenta ribbon format with the methyl-containing residues highlighted: 11 isoleucines (green), 22 leucines (blue), and 11 valines (orange). The associated nanodisc is shown in gray. The myristoyl chain (red) is shown as a ball and stick representation and expanded in the inset. Residues 2–13 form the N-terminal helix (embedded in the nanodisc), and residues 17–181 constitute the G-domain (solvent-exposed). For leucine and valine residues, only the Pro-S methyl carbons are shown. Images created using Chimera[75]. Panels (**B–D**) show $^1$H-$^{15}$N TROSY-HSQC, $^1$H-$^{13}$C HMQC, and $^1$H-$^{15}$N HSQC spectra of 100 μM $^2$H,$^{15}$N, $^{13}$CH$_3$-ILV myr-Arf1-GTP on a nanodisc (PC:PI(4,5)P$_2$ = 95:5), respectively.

membranes (stBLMs) in buffer containing 150 mM NaCl and 1 mM MgCl₂. PI(4,5)P₂ was included because it could bind to Arf[29–31] and is critical for ASAP1 ArfGAP activity[12,32]. stBLMs are solid-supported model lipid bilayers with physical properties similar to those of fully solvent-exposed membranes in vesicles or nanodiscs and are fully accessible to adsorbents from the adjacent buffer[33]. The latter was exploited to exchange GDP for GTP on Arf in situ, as described in "Methods".

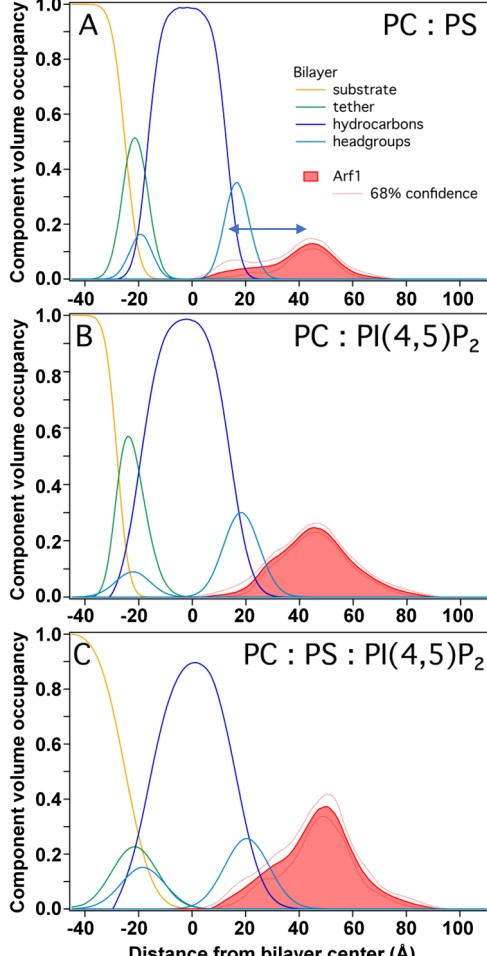

**Fig. 2 | Neutron reflectometry.** Component volume occupancy (CVO) profiles of membrane-associated myr-Arf1 and constituents of sparsely-tethered lipid bilayer membranes (stBLMs) composed of **A** PC:PS (85:15), **B** PC:PI(4,5)P₂ (95:5), and **C** PC:PS:PI(4,5)P₂ (87:10:3). The membrane coverage of Arf1 varies with the lipid composition. The shapes of the Arf1 profiles are similar: Arf1 associates peripherally with the lipid membrane, is primarily located in the membrane-adjacent bulk water phase, and the CVO profile peaks at ~45 Å from the bilayer center. Reflectivity curves and fit parameters for all stBLMs are provided in the Supplementary Information (Supplementary Figs. 4–6, Supplementary Table 1). Source data are provided as a Source data file.

Figure 2 contains the stBLM and myr-Arf1●GTP CVO profiles for all stBLMs. It shows that an increase in charge density of the stBLM, by adding PI(4,5)P₂, led to an increase in membrane coverage of Arf1 (Table 1), consistent with an electrostatic contribution of the N-terminal helix of myr-Arf1 to the membrane affinity[34], while the shape of the Arf1 CVO profiles remained similar. Arf is peripherally bound to the lipid membrane, with most proteinaceous material located outside the lipid membrane. The protein densities peak at ~28 Å from the center of the lipid headgroups and have a full-width-at-half-maximum (FWHM) of ~26 Å. Given the dimensions of the Arf1 G domain, this implies that the G domain, on average, is positioned above the membrane by ~5 Å. Arf1 CVO profiles at PI(4,5)P₂-containing stBLMs exhibit a shoulder towards the membrane interface that indicates a fraction of Arf1 configurations in which the G domain is in close contact with the lipid membrane. Therefore, the NR data support a dynamic ensemble of configurations of Arf1 at the membrane, where most configurations have the G domain somewhat displaced from the membrane.

### Accessibility of EBRs depends on G domain orientation

The NMR and NR results suggest a dynamic exchange between multiple states of the membrane-bound myr-Arf1●GTP. MD simulations naturally provide an avenue to obtain high-resolution information on the ensemble of conformations. Our approach is to capture unbiased membrane-partitioning and extensive sampling of lipid–protein interactions using atomistic MD simulations. We first performed nine independent membrane partitioning simulations of myr-Arf1●GTP with bilayers composed of (i) the zwitterionic PC headgroup alone, (ii) a binary lipid composition of PC:PI(4,5)P₂ (95:5), and (iii) a ternary lipid composition of PC:PS:PI(4,5)P₂ (80:15:5) (Fig. 3, Supplementary Table 2). PS was included to increase the anionic background charge while keeping the PI(4,5)P₂ concentration constant. To accelerate the membrane-partitioning process, we employed the highly mobile membrane mimetic (HMMM) in our membrane-binding simulations[35], which accelerates lipid diffusion and reorganization in an approximate model membrane, followed by full-membrane simulation, which included the full representation of POPC, POPS, and PI(4,5)P₂ lipid chains.

Simulations with bilayers containing only zwitterionic POPC showed shallow partitioning of Arf1 at the bilayer interface and a broad distribution of configurations located relatively far from the lipid bilayer (Fig. 3B, C; Supplementary Figs. 7, 10 and 11). Conversely, in simulations containing negatively charged lipids, the N-terminal helix was consistently positioned below the phosphate plane of the bilayer, and the myristoyl chain was inserted into the lipid bilayer (Fig. 3D, E; Supplementary Figs. 8, 9), in good agreement with lower off rate measured when bound to negatively charged membranes[34]. Interestingly, we observed similar $C_\alpha$ profiles for the membranes containing PI(4,5)P₂ only or PI(4,5)P₂ and PS lipids, suggesting that the conformational space explored by the G domain is not profoundly affected by the presence of bulk PS lipids. The calculated G domain center of mass relative to lipid headgroups (COM) confirmed that bulk PS lipids do not significantly affect the average position of the G domain ($COM_{POPC-POPS-PI(4,5)P2} = 22.9 \pm 1.8$ Å and $COM_{POPC-PI(4,5)P2} = 25.0 \pm 4.5$ Å)

## Table 1 | Arf1 G domain spatial distributions by NR

| | 85:15 POPC:POPS | 95:5 POPC:PI(4,5)P₂ | 87:10:3 POPC:POPS:PI(4,5)P₂ |
|---|---|---|---|
| Volume surface coverage of Arf1 (Å³/Å²) | 3.7 ± 0.6 | 8.2 ± 0.5 | 11.7 ± 0.5 |
| CVO peak distance from the center of the lipid bilayer (headgroups) (Å) | 45 ± 2 (28 ± 2) | 46 ± 2 (27 ± 2) | 49 ± 2 (29 ± 2) |
| CVO distribution FWHM (Å) | 21 ± 7 | 32 ± 3 | 26 ± 7 |

Amount, position, and distribution width of Arf1 from their component volume occupancy (CVO) profiles determined by NR using lipid bilayers of different composition. Values are derived from fits to the NR data (Supplementary Table 1) and uncertainties represent 68% confidence limits.

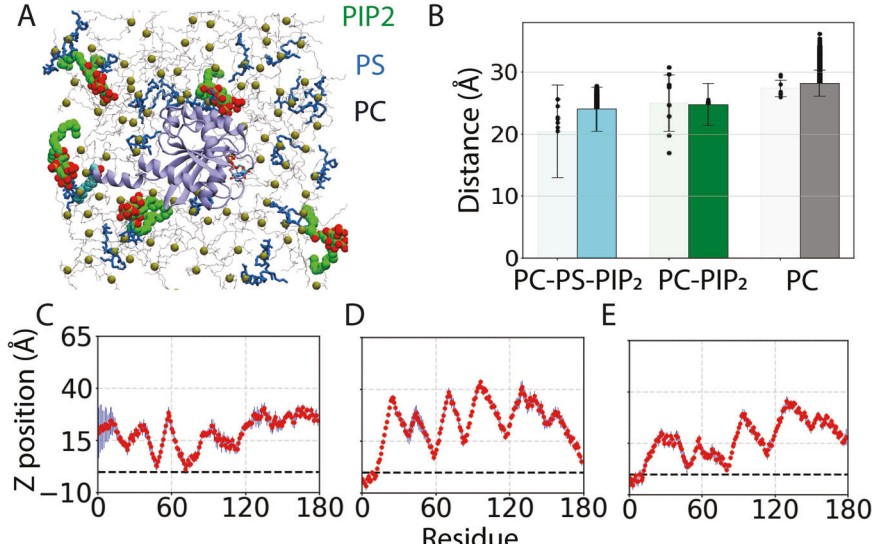

**Fig. 3 | Dynamics of the G domain in membrane-bound Arf1. A** Top view of a representative Arf1 conformation in a lipid bilayer containing PC, PS, and PI(4,5)P2 lipids, highlighted in different colors. **B** Average COM distance of the G domain (residues 17–181) from the lipid-bilayer (phosphorous plane set at 0), captured from multiple (9 ×100 ns) independent HMMM (light bars) and representative (1 × 1000 ns) full-membrane (dark bars) simulations in PC-PS-PI(4,5)P$_2$, PC-PI(4,5)P$_2$, and PC membranes. Standard deviation ($\sigma$) is calculated over the simulation trajectories and error bars represent +/−$\sigma$. Average COM distances for the representative replicas are highlighted in black dots. **C–E** Representative C$\alpha$ membrane insertion profile of Arf1, averaged over the last 50 ns of the nine independent HMMM membrane-binding replicas for **C** pure PC, **D** PC-PI(4,5)P$_2$, and **E** PC-PS-PI(4,5)P$_2$ membranes. The dashed line in each graph represents the position of the cis phosphorus plane (phosphorus atoms of the upper leaflet on which Arf1 was initially placed). Standard deviations are shown as blue bars. Source data are provided as a Source data file.

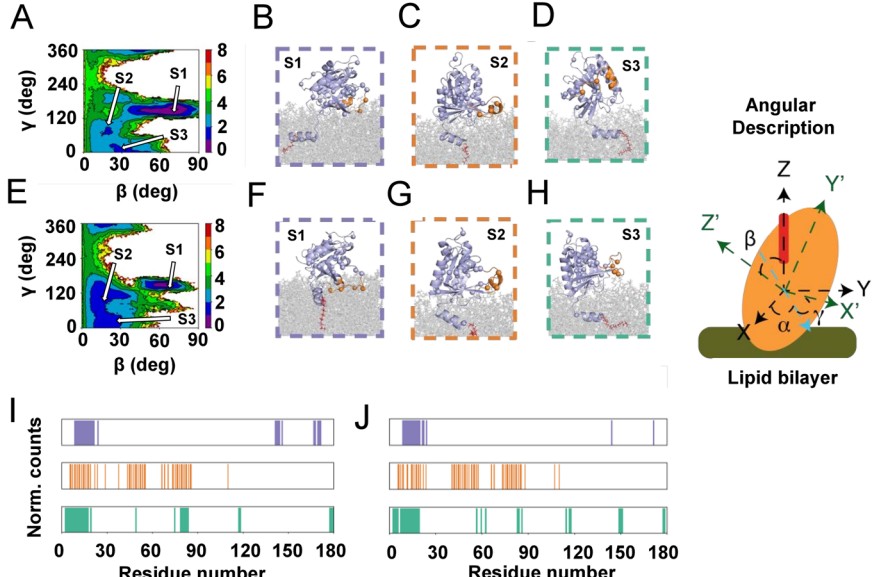

**Fig. 4 | Differential roles of PS and PI(4,5)P$_2$ lipids in orientational dynamics of Arf1 expressed in angular phase space (shown on the right). A** In the presence of PC-PS-PI(4,5)P$_2$-containing membranes, three distinct conformational states of Arf1 (S1, S2, and S3, highlighted in color boxes) are populated. **B–D** Representative snapshots of the membrane-bound Arf1 conformations corresponding to the three distinct states. **E** In the presence of PC-PI(4,5)P$_2$-containing membranes, three distinct conformational states are populated. For PC-PS-PI(4,5)P$_2$-containing membranes, the energy basins were more diffuse. **F–H** Representative snapshots of the PC-PS-PI(4,5)P$_2$ membrane-bound Arf1 conformations corresponding to the three distinct states. Methyl groups used in the NMR PRE analyses are shown as spheres, Arf1 is shown in cartoon representation, and helix 5 is colored in orange. Differential lipid–protein interactions (color-coded for states S1, S2, and S3 as in **B–D**) were captured in three distinct conformations in the presence of **I** PC-PS-PI(4,5)P$_2$ and **J** PC-PI(4,5)P$_2$ membranes. Source data are provided as a Source data file.

(Fig. 3B). In contrast, for a pure POPC membrane, the G domain localizes at a relatively farther distance (COM$_{POPC}$ = 27.4 ± 1.4 Å) from the lipid headgroups (Fig. 3B).

To better sample the conformational dynamics of the G domain, membrane-bound Arf1 HMMM simulations were converted to full-tailed lipid membranes and simulated for an additional 1000 ns

(aggregate sampling of 10 µs in each lipid environment). To decompose the conformational dynamics of Arf1 these multi-microsecond long simulations were projected onto a multi-dimensional collective variables space (CVs, see Supplementary Methods) (Fig. 4). The calculated potential of mean force (PMF), as a function of the orientation of the G domain, for bilayers with and without PS lipids revealed three

energetic minima for Arf1 in both lipid compositions (Fig. 4A, E). The minima correspond to three different rotational states, S1, S2, and S3, associated with differential interactions of switches I and II with the lipid bilayers (Fig. 4B, D, F–J). In the S1 cluster ($\beta = 57$–$73°$, $\gamma = 115$–$187°$), switch I is buried within the lipid bilayer and switch II is localized at the membrane interface such that we can refer to S1 as an "occluded" conformation. In the S3 cluster ($\beta = 20$–$30°$, $\gamma = 0$–$50°$), switches I and II are entirely solvent exposed, while the S2 cluster ($\beta = 8$–$20°$, $\gamma = 64$–$118°$) is characterized by an exposed switch I and an interfacial switch II (Fig. 4I, J). Interestingly in the presence of only zwitterionic PC, the PMF profile captured a broad minimum where the G domain was not in intimate contact with the lipid bilayer (Supplementary Figs. 10, 11).

## Combined NMR-PRE and MD analysis of G domain conformations

The MD results suggest a dynamic equilibrium between multiple states of the membrane-bound myr-Arf1•GTP, in which functionally relevant parts of the G domain are either exposed or occluded via interactions with the lipid bilayer. To experimentally probe this conformational distribution, bilayer proximities of the I, L, and V methyl groups of myr-Arf1•GTP were estimated from paramagnetic relaxation enhancement (PRE) NMR data[36]. These experiments measure the ratios of residue-specific spectral intensities, $I/I_0$, in the presence and absence of spin-labeled lipids in the bilayer. Side-chain methyl groups provide excellent reporters in such high molecular weight assemblies, where high-quality spectra are observable for mobile and fixed domains relative to the membrane (Fig. 1C)[16,37]. Furthermore, Arf1 contains 11 I, 22 L, and 11 V residues, which are well dispersed throughout the entire sequence. Structurally, one I (I4) and two L (L8, L12) residues are in the N-terminal helix, and the remaining forty-one residues are widely distributed throughout the G domain, including the functionally relevant sw1 (I42, V43, I46, and I49) and sw2 (I74, L77). Thus, the methyl groups provide balanced reporters for membrane proximity.

Nanodiscs containing PI(4,5)P$_2$ or PI(4,5)P$_2$ and PS were prepared with and without 5-doxyl PC. The molar fraction of 5-doxyl PC was chosen to have, at minimum, one spin-labeled lipid per monolayer after myr-Arf1 nucleotide exchange. Figure 5A shows the resulting PREs, plotted as $I/I_0$ for each of the I, L, and V side-chain methyl groups mapped on the structure (Fig. 5B). The N-terminal helix residues I4, L8, and L12 have very low $I/I_0$, consistent with the helix being buried in the bilayer surface[3,38]. The PRE for a given conformation of the G domain may be computed as outlined in the Supplementary Methods. The multiple extended (1 μs) full-membrane MD trajectories provide a conformational ensemble that may be used to analyze the PRE data through back calculation of the expected PRE for each conformation. Analysis of these conformations reveals that it is impossible to fit the PRE data to any single conformation of the G domain relative to the membrane (Supplementary Fig. 12). This observation is consistent with the NMR spectral analysis of flexibility and correlation times (see above). Furthermore, the data do not fit an average of all conformations observed in the MD trajectories (Supplementary Fig. 13) Hence, we assume fast exchange between the conformations and take an ensemble approach (see Supplementary Information) to select the minimal set of conformations required to fit the observed PRE data. The ensembles were selected from a set of conformations reported in the MD simulations, based on each conformation's theoretical PRE response (see Supplementary Methods). An ensemble size of 16 was used to fit the experimental PRE profiles (Supplementary Figs. 12–15) obtained for POPC nanodiscs containing PI(4,5)P$_2$ (Fig. 5C) or PI(4,5)P$_2$ and POPS (Fig. 5G), which is consistent with a mobile distribution of the G domain above the membrane surface. On the average, the COM of the G domain is found to be separated from the membrane surface by 23 Å in POPC/PI(4,5)P$_2$ nanodiscs and by 21 Å in POPC/POPS/PI(4,5)P$_2$ nanodiscs. Further, these analyses reveal three states that back

predict the observed PRE and qualitatively resemble the distribution observed in the MD simulations (Figs. 4B–D, F–H, 5D, 5H). However, the MD simulations suggest a larger population of conformers in the S1 state than the PRE ensemble analysis selects to fit the experimental data. The observed discrepancies between the MD simulations and PRE-based ensemble selection might arise due to force field limitations in accurately capturing non-covalent interactions. A composite PRE profile for the each of the S1, S2, S3 conformation ensembles are shown in Supplementary Fig. 12C, D. We observed that the relative population of the S1 state increases for nanodiscs containing POPC/POPS/PI(4,5)P$_2$ lipids compared to nanodiscs containing POPC/PI(4,5)P$_2$ lipids (Fig. 5F, J), suggesting that charge-charge interactions between POPS head-groups and the G domain may increase this population. For the ensembles, the average membrane separation of $COM_{POPC-PI(4,5)P2}$ (23 Å) decreases slightly by the presence of POPS in the membrane, $COM_{POPC-POPS-PI(4,5)P2}$ (22 Å). Such putative interactions are indicated in the MD simulations (Supplementary Figs. 16, 17). However, it is essential to note that we have no independent experimental validation of these contacts, and the difference in the experimental PRE curves is relatively small (on the order of the experimental errors).

## Myristoylated N-terminal α helix anchors Arf1 to membrane

Details on the association of the myristoylated N-terminal helix of Arf1 with the membrane were obtained by MD simulations and solid-state $^2$H NMR. Analysis of the full-membrane simulations shows that the N-terminal helix comprising residues 2–15 is oriented perpendicular to the membrane normal (angle = $93.5° \pm 12.1°$) and inserted on average $6.5 \pm 2.3$ Å below the phosphate plane of the bilayers containing anionic lipids. Furthermore, the myristoyl chain extends into the lipid matrix.

In solid-state $^2$H NMR experiments of myr-Arf1 in POPC model membranes, the myristoyl chain of myr-Arf1 and the acyl chains of the POPC membranes were investigated using complementary samples: (1) POPC-$d_{31}$, (2) POPC-$d_{31}$ + myr-Arf1, and (3) POPC + $d_{27}$-myr-Arf1. The $^2$H NMR spectra (Fig. 6A) have the expected shape and resolution for POPC membranes. In such spectra, Pake-doublets are observed for each labeled position of the hydrocarbon chain. These Pake-doublets are symmetric line shapes that represent all possible orientations of a C–D bond vector in a powder-type sample and the distance between the two most intense peaks of the powder pattern can be directly related to the segmental order parameter, which describes the amplitude of the motion of the corresponding C–D bond. In Fig. 6A, resolved Pake-doublets are visible that can be assigned to individual positions in the palmitoyl chain in sn-1 position of POPC[39]. Adding myr-Arf1 to the POPC membranes led to only minor changes in the spectral shape (Fig. 6A). In contrast, the $^2$H NMR spectrum of the $d_{27}$-myr-Arf1sample differs considerably from the POPC-$d_{31}$ spectra. The maximum width is very similar, but in contrast to POPC-$d_{31}$, the spectral intensity significantly increases towards the center of the spectrum. Also, the quadrupolar splitting of the terminal CH$_3$ group is much narrower. These data indicate that the myristoyl chain of Arf1 is, on average, less ordered than the palmitoyl chain of POPC.

To further quantify the reduced order of the myristoyl chain, smoothed chain order parameter profiles were calculated from the NMR spectra (Fig. 6B). The order parameters (S) represent a measure of the amplitude of motion of a C-D segment, wherein a smaller order parameter corresponds to larger amplitude motions. The difference in the order parameters obtained for the palmitoyl chain of POPC in the absence and presence of Arf1 is very small and close to the experimental error, indicating no perturbation of the lipid packing by Arf1 binding. In contrast, the order-parameter profile of the Arf1 myristoyl chain shows much lower order in general. It only matches the order parameters of POPC in the uppermost part of the chain. To further interpret the order parameters concerning chain geometry, we used an analytical model to calculate chain extension profiles that contain the

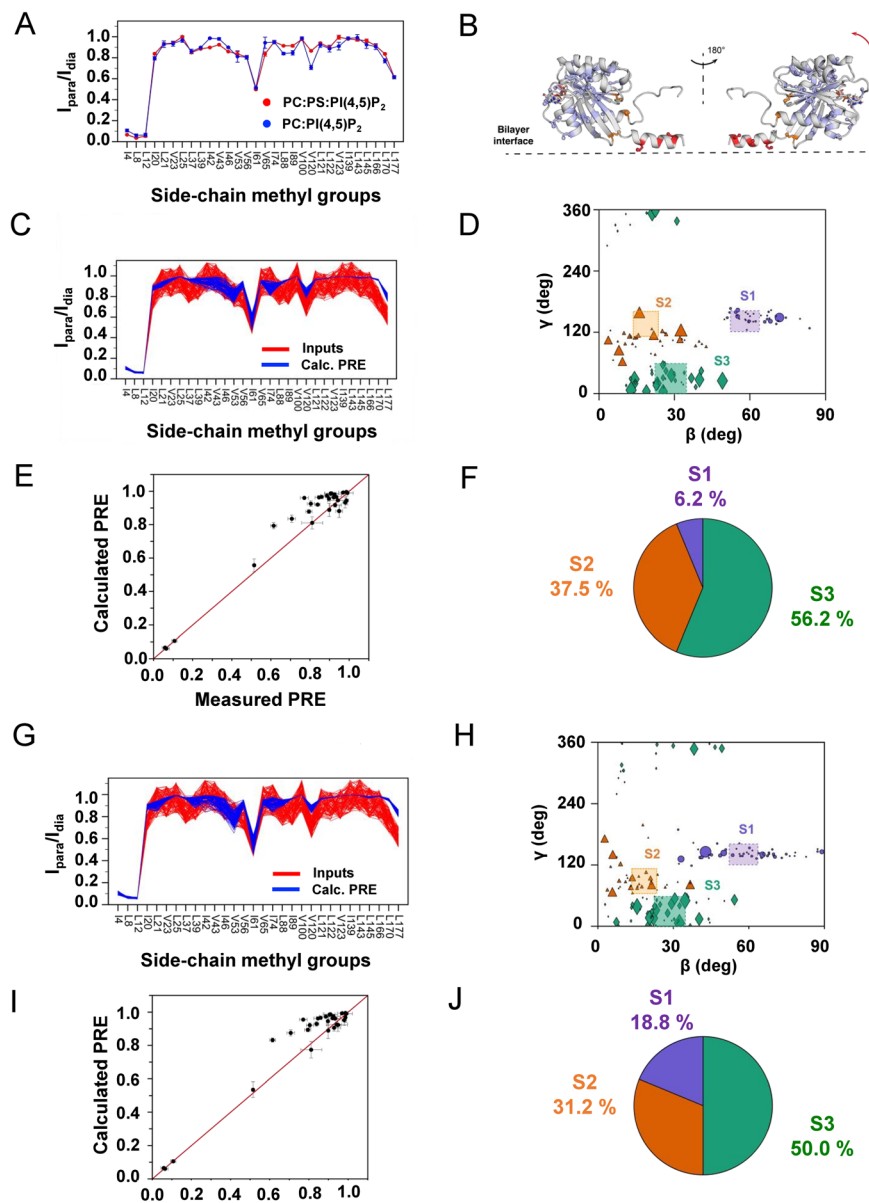

**Fig. 5 | PRE-driven conformational analysis of the G domain. A** PRE profiles measured on a POPC/POPS/PI(4,5)P$_2$ (75/20/5) membrane (red) and on a POPC/PI(4,5)P$_2$ (95/5) membrane (blue) for myr-Arf1•GTP anchored to nanodiscs doped with 5 mol% spin-labeled PC (5-doxyl PC) to replace POPC. Two independent experiments were performed. Data are presented as mean values. Error bars were calculated based on the signal-to-noise (S/N) ratio of the spectra as described in Methods. **B** PRE effects mapped on the myr-Arf1•GTP structure, based on measurements from a PC/PI(4,5)P$_2$ membrane. The side chains of isoleucine, leucine, and valine residues are shown in stick representation, with the methyl carbons as spheres. Red highlighted spheres are residues I4, L8, and L12 on the N-terminal α-helix with significant PREs. Orange spheres indicate residues I61, V120, and L177 with intermediate PRE effects. The remaining blue spheres represent methyl groups with weak or no PREs. Panels (**C–F**) apply to PC-PI(4,5)P$_2$ membranes, and panels (**G–J**) apply to PC-PS-PI(4,5)P$_2$ membranes. **C, G** Comparison of experimental PRE data with 15% random error (red) with the best-fit 16-member ensemble (blue).

symbol indicates the relative number of times this conformation appears in the conglomerate 100 computations. Conformations are clustered in ranges identified in the MD analyses (Fig. 4 and Supplementary Information), referred to as S1 (magenta), S2 (orange), and S3 (green). **E, I** Correlation of the calculated average PRE versus measured PRE for the best-fit 16-member subset of structures (from the set of MD conformations), corresponding to (**D, H**). For the measured PRE (x-axis), error bars were calculated based on the spectral signal-to-noise ratio as described in Methods. For the calculated PRE (y-axis), data are presented as mean values +/−SD. **F, J** Populations of S1, S2, and S3 from the best-fit ensemble (**F**: PC:PI(4,5)P$_2$ S1 = 6.2% +/− 4.2 (magenta); S2 = 37.5% +/− 10 (orange); and S3 = 56.2% +/−9.7 (green). **J**: PC:PS:PI(4,5)P$_2$ S1 = 18.8% +/− 5.7 (magenta); S2 = 31.2% +/− 9.7 (orange); and S3 = 50% +/− 9.5 (green)). Data are presented as mean values +/−SD. Error bars were calculated based on 100 repeats of ensemble analysis. Statistical significance was assessed by two-sided Student's *t* test, giving *p* < 0.0001. Source data are provided as a Source data file.

distances of individual carbon positions in the hydrocarbon chain from the terminal CH$_3$ group projected on the membrane normal[40] (Fig. 6C). The terminal CH$_3$ groups were arbitrarily placed at a chain extension position of zero, which does not imply that the CH$_3$ groups of the POPC lipids and the myristoyl chain are at the same depth in the membrane.

Instead, the profiles could be shifted up and down relative to each other. Therefore, to reliably compare the chain extension profiles for two different molecules, its slope should be considered. In this case, the slope of the chain extension profiles of the myr-Arf1 myristoyl chain is visibly smaller than that of the palmitoyl chain of POPC. This

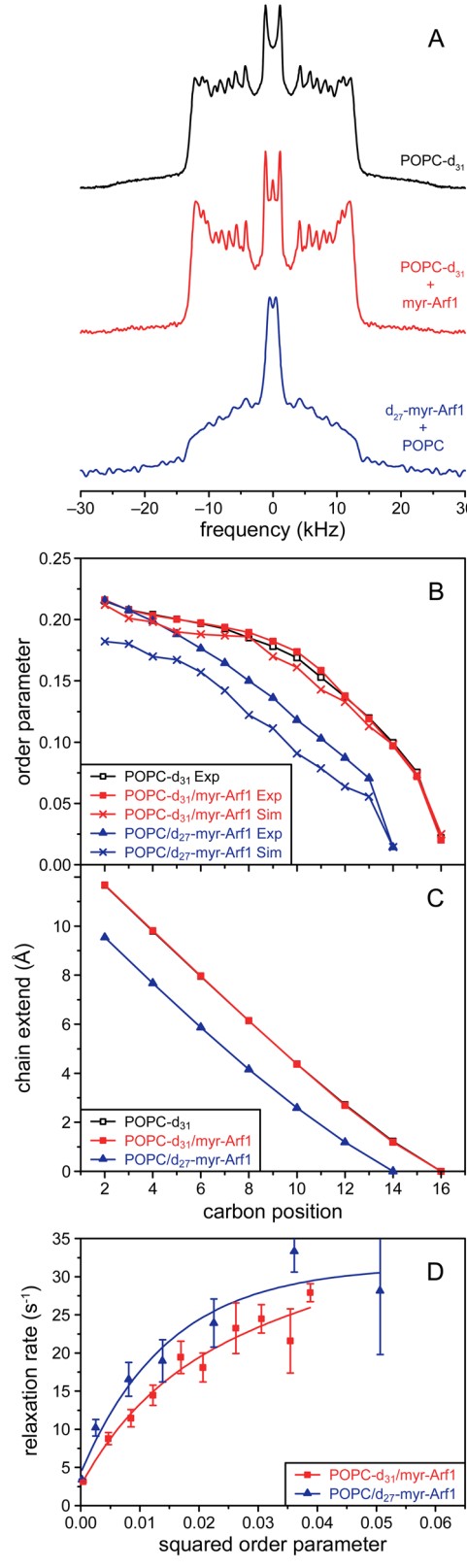

**Fig. 6 | Embedding of the myristoyl chain of myr-Arf1 in the membrane bilayer.** In all panels data for POPC-d$_{31}$ is shown in black, data for POPC-d$_{31}$+myr-Arf1 in red, and data for d$_{27}$-myr-Arf1+POPC in blue. Data points for experimental results are shown as either square (POPC deuterated) or triangle (myr-Arf1 deuterated) symbols while the corresponding data points from MD simulations are shown as cross symbols. **A** $^2$H NMR spectra and **B** corresponding smoothed order parameter profiles were obtained for samples of pure POPC bilayers or bilayers containing myr-Arf1 at a lipid to protein ratio of 150:1 and from simulations. **C** Chain extension profiles were obtained from smoothed order parameter profiles in (**B**). **D** Plot of R$_{1Z}$ versus S$^2$ for the individual carbon positions in the POPC-d$_{31}$ chain and the myr-d$_{27}$ chain. Lines are drawn to guide the eye. Error bars represent the 95% confidence interval of the relaxation rates obtained from fitting simulated spectra to the experimental spectra. Source data are provided as a Source data file.

data indicate that the myristoyl chain is fully embedded and extended in the membrane, thus providing additional anchoring for the N-terminal helix of myr-Arf1.

Analysis of the MD trajectories permits the computation of the order parameters for both the myristoyl chain of myr-Arf1 and the acyl chain of the POPC lipid. These data are also shown in Fig. 6B and agree relatively well with the experimental ssNMR data.

### Binding ASAP1–PH domain alters G domain dynamic equilibrium

The ArfGAP ASAP1 has a core catalytic domain composed of PH, Arf-GAP, and ankyrin repeat domains ([325-724]–ASAP1, referred to as PZA (for PH, Zinc binding, which comprises the ArfGAP catalytic domain, and Ankyrin repeat domains)). Recently, we showed that binding of the ASAP1 PH domain to Arf1●GTP at the membrane surface might participate in regulation of GTP hydrolysis[5,16]. To probe the influence of that interaction on the dynamic equilibrium of the G domain, we recorded $^1$H-$^{15}$N TROSY spectra of nanodisc-bound [$^2$H, $^{15}$N]-myr-Arf1●GTP in the presence of unlabeled ASAP1-PH (Fig. 7A). Surprisingly, the addition of a stoichiometric ratio of ASAP1-PH resulted in the loss of nearly all amide backbone resonances in the TROSY spectrum, in stark contrast with the spectrum in the absence of ArfGAP (Fig. 1B). Since it is known that ASAP1-PH will bind to the membrane surface via PI(4,5)P$_2$ and that $^{L8K}$Arf1●GTP binds to PH[16], the loss of $^1$H-$^{15}$N NMR signals suggests that the G domain binds to PH at the membrane surface and becomes locked, reorienting with the same correlation time as the nanodisc. To confirm that the disappearance of the spectrum was due to a change in orientational dynamics and not an exchange between conformations at an intermediate time scale, we recorded $^1$H-$^{15}$N chemical shift correlation spectra using an HSQC pulse sequence, which allows the observation of side-chain amine groups in addition to backbone amide resonances (Fig. 7B). The cross peaks expected for side-chain amine groups were still observable, while the backbone amides remained unobservable, indicating that the additional side-chain motions lengthen the relaxation times and allow detection even as the correlation time of the whole complex has increased. These observations confirmed that the binding of the ASAP1 PH domain did indeed alter the conformational space explored by the Arf G domain.

## Discussion

Lipidated small GTPases, such as Arf1, bind to membranes, where they orchestrate various regulatory functions of lipid and membrane trafficking by interacting with multiple effectors and regulators. Defining Arf1 membrane-bound states is therefore fundamental to understanding how a single protein can interact with disparate effectors. By using a combination of biophysical measurements and MD simulations, we describe (i) the details of Arf1 anchoring at the membrane by its myristoylated N-term helix, (ii) how the Arf1 G domain is characterized by a dynamic equilibrium between multiple functionally relevant states, and (iii) how the G domain dynamics are altered by its interaction with the ASAP1 PH domain.

indicates that the myristoyl chain of Arf1 is immersed in the bilayer and is somewhat compressed along the membrane normal and therefore exhibits greater lateral mobility compared to the palmitoyl chain of POPC. The higher mobility is also evident in the relaxation behavior of the two chains (Fig. 6D). The curvature of the R1Z vs. S$^2$ plots indicates that the myristoyl chain of myr-Arf1 is considerably more flexible than the surrounding palmitoyl chains of POPC lipids[41,42]. In summary, these

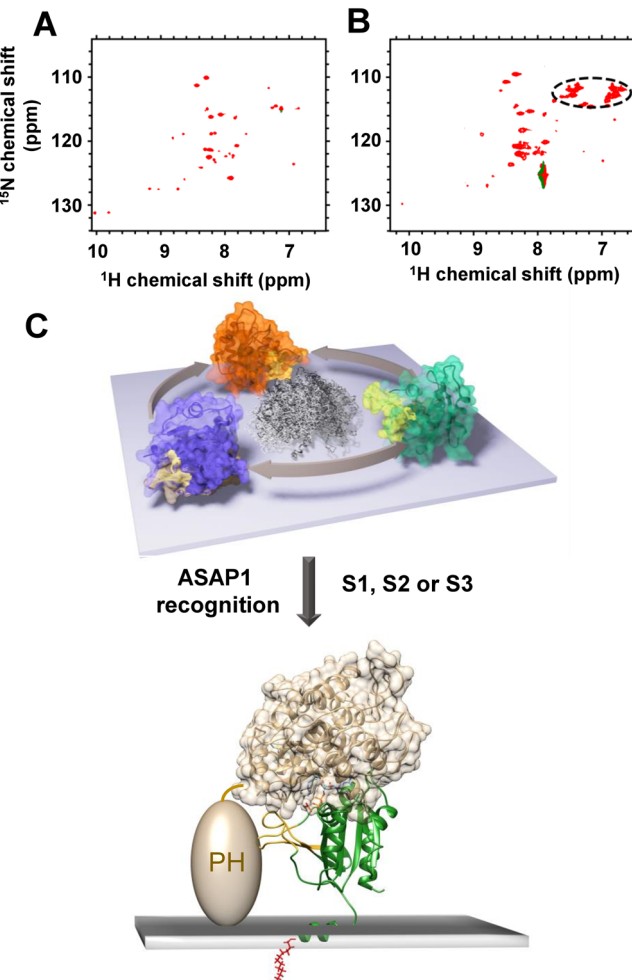

**Fig. 7 | myr-Arf1 interaction with its effectors and the basis of conformational selection.** 2D $^1$H-$^{15}$N spectra of myr-Arf1•GTP in the presence of 100 μM ASAP1 PH (myr-Arf1-GTP:ASAP1 PH = 1:1), shown as **A** $^1$H-$^{15}$N TROSY-HSQC and **B** $^1$H-$^{15}$N HSQC spectra. Asparagine and glutamine side-chain amine groups are circled. **C** Arf1 G domain presents multiple states (center ensemble) to recognize signaling partners. NMR and MD analyses identify three populated conformations (S1 (blue), S2 (orange), S3 (green) with switch 1-switch 2 highlighted in yellow) that mutually exchange on the surface of the membrane. Postulated recognition of the S3 conformation by ASAP1 PZA is modeled based on the structure of the Arf6:ASAP3 GAP complex[48] (ASAP3 shown as tan ribbon and surface based on PDB: 3LVQ; S3 state of myr-Arf1 in the membrane shown as green ribbons with switch 1, switch 2 in yellow ribbon and the myr-chain in red). A putative position of the ASAP1 PH domain (tan colored ellipsoid, positioned N-terminal to the GAP and ankyrin domains), which interacts with the membrane surface[16], suggests how binding to PH at the membrane would form a motionally restricted complex (panels **A** and **B**). Images created using Chimera[75].

The association of myr-Arf1 with the membrane is characterized by (i) embedding of the N-terminal α-helix into the membrane surface (Figs. 3–5, Supplementary Fig. 17), and (ii) full integration of the N-terminal myristoyl chain into the membrane (Fig. 6). Combined, the solution NMR PRE and MD results indicate that the N-terminal α-helix is oriented almost parallel to the membrane surface with hydrophobic sidechains inserted in the bilayer and embedded with the helical axis ~6.5 Å below the phosphate plane of the bilayer (Fig. 3D, E). The orientation results are in agreement with prior peptide studies[7–9]; however, no embedded depth was determined in earlier studies. Results from solid-state NMR indicate that the myristoyl chain of myr-Arf1 is fully inserted into the membrane matrix and is slightly more flexible and disordered than the palmitoyl chain of the surrounding

POPC lipids. Earlier studies suggested that the myristoyl chain folded back along the N-term helix at the membrane surface[3]; however, this is inconsistent with the present NMR and MD data. It is possible that this result stemmed from use of a bicelle system or reflects transient mobility of the chain detected by NOE studies[3]. Computed order parameters for the myristoyl chain from the MD simulations agree with the average experimental order parameters from solid-state NMR, supporting the chain insertion model. The combined effects of the embedded α-helix and myristoyl chain insertion are consistent with the stable association of myr-Arf1 with the membrane, in contrast to other systems, wherein a single lipid modification, despite its considerable contribution to the binding enthalpy, is usually insufficient for stable anchoring of peripheral membrane proteins due to the associated entropy cost of immobilizing the protein at the membrane interface[43,44]. In other systems, a second contribution to membrane binding is needed, which can either be additional lipid modifications, as for H-Ras or N-Ras and many other proteins[45], or electrostatic interactions of charged amino acids in the anchor, as for K-Ras, MARCKS, Src, or HIV-1 Gag[46]. In the case of Arf1, this additional contribution stems from the immersion of the N-terminal α-helix in the lipid-water interface of the membrane and interaction with charged lipids[6,30,31,34]. This arrangement is supported by embedding hydrophobic F, L, I, and A residues of the N-terminal α-helix below the membrane surface and the presence of charged (K) and polar (N) residues at the membrane surface (Supplementary Figs. 16, 17). We note that this mechanism of membrane anchoring differs considerably from the doubly lipid-modified N-Ras protein, where a non-hydrogen bonded protein backbone is associated with the membrane interface and only the hydrophobic sidechains point towards the membrane interior[47].

In terms of lipid chain anchoring, it has been shown that the lipid modifications of N-Ras match the length of the surrounding membrane matrix, such that the chains always reach the central plane, where the two monolayers of the membrane meet[47]. Assuming that the same principle is true for Arf1, the shorter length of the myristoyl chain would suggest that the N-terminal α-helix is embedded into the membrane to compensate for the different chain length compared to the surrounding lipids and results in the overall reduced order of the chain (Fig. 6B, C). Mathematical models of acyl chain geometry[40] show that low-order parameters of acyl chains are associated with their relatively large cross-sectional areas. In the present case of myr-Arf1, the observation that the N-terminal α-helix is embedded in the lipid-water interface, where it occupies a large amount of interfacial area, is consistent with the myristoyl chain underneath the α-helix having more motional freedom and exhibiting lower order parameters than the adjacent lipids. The square-law plots of $R_{1Z}$ vs. $^2$H-order parameters (Fig. 6D) show that the myristoyl chain is more flexible than the palmitoyl chains of POPC, and fills the space below the α-helix. In such plots, stiffer membranes (as experienced in the presence of cholesterol) result in straight lines with a very shallow slope, while flexible membranes (as experienced in the presence of detergents) result in bent plots with steep slopes[41,42]. The plots for the myristoyl chain of Arf1 and the palmitoyl chain of POPC look relatively similar, indicating highly flexible and elastic packed membranes as also experienced for saturated membranes in the presence of detergents. It is possible that the reduced order parameters infer sufficient mobility to explain transient NOE connectivities seen previously[3], while still supporting the full chain insertion. Collectively, these observations provide an explanation for the tight affinity of myr-Arf1 for membranes with only a single lipid modification.

Signaling by Arf1 is mediated by the G domain, when myr-Arf1•GTP is anchored to the membrane. The solution NMR, NR, and MD data indicate that the Arf1 G domain is quite mobile above the membrane surface of nanodiscs. The G domain has a correlation time of 34 ns compared to the expected correlation time for a rigid

Arf1:nanodisc complex of 60 ns. Collectively, these data suggest that the G domain must oscillate about the short linker between the N-terminal α-helix and the G domain. This behavior is similar to previous reports for yeast myr-Arf1 bound to bicelles[3] and the related K-Ras GTPase protein[22,23], which has a 19-residue C-terminal hypervariable region, terminating with a farnesyl group serving as the membrane tether that allows more flexible, faster, and less-restricted mobility above the membrane surface ($\tau_c$ ~ 20 ns). The motion of K-Ras GTPase was examined using a similar methodology as presented here[22,23] and was analyzed to yield two conformational states, occluded and exposed. Based on the observed faster correlation time for the G domain of K-Ras$_{4B}$, these two states are most likely a fast average over a distribution of states. Arf1's linker between the N-terminal amphipathic helix and the G domain is much shorter, suggesting a restricted conformational space. However, the rotational energy barriers between conformations of myr-Arf1 are low.

The NMR, MD, and NR data collectively reveal that the G domain conformational space populates states (S1, S2, and S3) that can present interfaces leading to productive complex formation with effectors (primarily via either S2 or S3) and functional signaling. Interestingly, the database of conformations from the MD trajectories contains a significant population of S1 conformations with occlusion of the switch 1 - switch 2 regions; however, the PRE back-calculation selection method only requires a very small population of these states to adequately fit the data. The vast majority of states are in the S2- and S3-type conformations, similar to the distal states revealed in the NR data.

The functional consequence of the three states may be envisioned by modeling a binding complex via superposition of the in vitro structure of the related Arf6 G domain and the GAP domain of ASAP3 (PDB:3LVQ)[48] with a membrane bound conformation of the Arf1 G domain from the S3 state (Fig. 7C). The model illustrates that the binding mode of myr-Arf1 to the membrane can be accommodated and present the putative binding surface for interaction with the GAP domain by selection of the S3 state from the dynamic landscape. For Arf1/ASAP1, we have shown that the ASAP1 PH domain recognizes PIP2 in the membrane and can associate with Arf1[16]; hence, it is reasonable to expect that recognition with the effector ASAP1 PH or ASAP1 PZA (containing the GAP domain) may lead to a single, immobilized, and well-defined complex as one proceeds along the functional reaction coordinate. The effector-Arf1 complex should then lose the observed G domain dynamics and reflect the molecular tumbling (correlation time) of the entire nanodisc complex (>60 ns). Upon addition of the PH domain of ASAP1 to myr-Arf1●GTP/PC:PI(4,5)P$_2$-nanodiscs, indeed, the G domain mobility was quenched, and $^1$H-$^{15}$N TROSY spectra were undetectable (Fig. 7A). The reduced dynamics of the Arf1 G domain induced by ASAP1 PH binding are caused by PH first binding to the PI(4,5)P$_2$ lipids[16] and subsequently binding to the membrane-anchored Arf1 G domain. The model (Fig. 7C) indicates that the ASAP1 PH domain could easily fit alongside the Arf1 G domain and further position or allosterically modify the Arf1 G domain to form the active ASAP1-PZA:Arf1 complex. These events are readily detectable using the planar membrane mimetic (nanodiscs) comprised of the appropriate lipid compositions. Future studies using these systems will examine the interfaces and functional consequences of Arf1/ASAP1 complexes.

Our studies illustrate the combined power of integrated computational and experimental methods to address significant biological problems that do not conform to traditional structural tools[49] and provide an in-depth analysis of the dynamics and conformational complexity of myr-Arf1 at the membrane surface. The presentation of different surfaces of Arf1 in separate states enables the recognition of different functional effectors, which have been previously shown to interact with different surfaces of the Arf GTPases. This multi-state complexity is consistent with the short linker and tight surface interactions of the Arf GTPases. This behavior contrasts with other GTPases, such as K-Ras, which have longer, more flexible linkers. It will be of interest to ascertain if mutations in Arf1 modify the multi-state dynamics or recognition of effectors, similar to observations for K-Ras[22]. These observations suggest classifications of the signal recognition mechanisms by membrane surface complexes of small GTPases. Understanding the configurational dynamics of these GTPases is of high functional relevance.

## Methods

### Construction of dual expression vector for myr-Arf1

The dual gene expression vector pETDuet-1 (Novagen) has two multiple cloning sites (MCSs) MCS1 and MCS2, each of which has a T7 promoter/*lac* operator and ribosome binding site. Two target genes were synthesized (Integrated DNA Technologies, Inc.) and subcloned into the respective MCSs. The human Arf1 (Uniprot P84077) gene was cloned into MCS1 of the pETDuet-1 vector between *Nco I* and *Sac I* restriction sites. A GSGSHHHHHH-tag was added at the C-terminus of human Arf1. The yeast NMT (Uniprot P14743) gene was subcloned into MCS2 of the pETDuet-1 vector between *Ndel I* and *Xho I* restriction sites.

### Expression and purification of U-[$^2$H],$^{13}$CH$_3$-methyl myr-Arf1

For expression of the myr-Arf1 protein (~21 kDa), the pETDuet-1 plasmid was transformed into *E. coli* BL21 Star (DE3) cells (Invitrogen), plated on LB agar plate containing carbenicillin (100 mg/L) for overnight growth. Expression of deuterated myr-Arf1 generally followed the recently published protocol[19], modified as follows. The freshly transformed colonies were then picked and resuspended into 10 mL of M9/H$_2$O media for overnight growth at 37 °C in a shaking incubator. Then the overnight culture was poured into a fresh M9/H$_2$O media with a total volume of 5 mL and OD$_{600}$ of 0.2 and continued to grow until an OD$_{600}$ of about 0.6. The culture was diluted one-to-one with M9/D$_2$O medium (prepared with D-[$^2$H;$^{12}$C]-glucose) and incubated until OD$_{600}$ reached 0.6. After repeating the same dilution procedure twice (with the final culture volume of 40 mL), the cells were spun down and resuspended in 200 mL M9/D$_2$O medium for overnight growth at 37 °C. The expression culture was made from the overnight culture by diluting to a volume of 2 L with a starting OD$_{600}$ of about 0.2. After the culture had reached an OD$_{600}$ of 0.6, the temperature was reduced from 37 °C to 22 °C. For the production of myr-Arf1, sodium myristate (Sigma-Aldrich, M8005) was added 10 minutes before induction to a final concentration of 100 μM. At the same time, the media was supplemented with: 1) 50 mg/L 2-keto-3-[D$_2$],4-[$^{13}$C]-butyrate (Cambridge Isotope Laboratories, Inc. CDLM-7318) and 100 mg/L 2-keto-3-[D]-[$^{13}$CH$_3$,$^{12}$CD$_3$]-isovalerate (Cambridge Isotope Laboratories, Inc. CDLM-7317) to enable selective labeling of ILV methyl groups; 2) 50 mg/L coenzyme A sodium salt (Sigma-Aldrich, C3144) to promote efficient N-myristoylation. Protein expression was induced at an OD$_{600}$ of 0.8 by adding isopropyl-β-D-thiogalactopyranoside (IPTG) to a final concentration of 0.2 mM. The culture was incubated for additional 16 h at 22 °C for protein expression. Cells were harvested by centrifugation at 7000 × *g*, 4 °C for 30 min.

The cell pellets were resuspended in 25 mL lysis buffer (20 mM Tris-HCl, pH 8.0, 150 mM NaCl, 20 mM imidazole, 1 mM MgCl$_2$, and 0.5 mM tris(2-carboxyethyl)phosphine (TCEP)) with one tablet of ethylenediaminetetraacetic acid (EDTA)-free protease inhibitor (Thermo Scientist, A32965). The cells were lysed with a model 110 S microfluidizer (Microfluidics) and clarified by centrifugation at 48,000 × *g* and 4 °C for 45 min. The lysate was loaded onto two 5 mL HisTrap HP columns (GE Healthcare). After the columns were washed with six column volumes (CVs) of lysis buffer, Arf1 and myr-Arf1 were eluted with an identical buffer containing 300 mM imidazole in a linear gradient from 20 mM to 300 mM imidazole over 14 CVs. The purity of myr-Arf1 was examined by LC-MS. The fractions containing purified myr-Arf1 were pooled and kept at 4 °C for further processing. The fractions containing both Arf1 and myr-Arf1 were combined and

concentrated to a volume of one milliliter. Sodium chloride crystals were added to the sample to a final concentration of 3 M. After centrifugation at 21,000 × g and 4 °C for 15 min, the supernatant was collected and applied to a 5 mL pre-equilibrated HiTrap Phenyl HP hydrophobic interaction column (GE Healthcare) using a running buffer with 20 mM Tris-HCl, pH 7.4, 3 M NaCl, 1 mM MgCl$_2$, and 0.5 mM TCEP. After the column was washed with ten column volumes of running buffer, myr-Arf1 was eluted with 20 mM Tris-HCl (pH 7.4), 1 mM MgCl$_2$, and 0.5 mM TCEP using a linear gradient. The purity of myr-Arf1 was confirmed by LC-MS. Purified myr-Arf1 was exchanged to a buffer condition of 20 mM Tris-HCl, pH 7.4, 150 mM NaCl, 1 mM MgCl$_2$, and 0.5 mM TCEP, concentrated to about 150 µM, and stored at −80 °C for further usages. The protein concentration was calculated by measuring the absorbance at 280 nm using a molar extinction coefficient of 29,450 M$^{-1}$ cm$^{-1}$.

### Preparation of myr-Arf1·GTPγS anchored on MSPΔH5 nanodiscs

All lipids were purchased from Avanti Polar Lipid, Inc. and were mixed in chloroform solution, then air-dried with dry nitrogen gas and resolubilized with cholate aqueous buffer (20 mM Tris-HCl, pH 7.4, 150 mM NaCl, and 75 mM sodium cholate). For the diamagnetic reference sample, NDs were prepared using the desired phospholipids of 1-palmitoyl-2-oleoyl-glycero-3-phosphocholine (POPC), 1-palmitoyl-2-oleoyl-sn-glycero-3-phospho-L-serine (POPS), and 1,2-dioleoyl-sn-glycero-3-phospho-(1′-myo-inositol-4′,5′-bisphosphate) (PI(4,5)P$_2$). For paramagnetic samples, 1-palmitoyl-2-stearoyl-(5-doxyl)-sn-glycero-3-phosphocholine (5-doxyl PC) were incorporated replacing a small fraction of POPC (5 mol%).

U-[$^2$H],$^{13}$CH$_3$-methyl labeled myr-Arf1 and unlabeled MSPΔH5 (in the molar ratio of 1:1) were incubated with cholate-resolubilized lipids in 20 mM Tris-HCl, pH 7.4, 150 mM NaCl, 0.5 mM MgCl$_2$, 0.5 mM TCEP, 1 mM EDTA, and 2 mM GTPγS with a final cholate concentration of 18 mM. After incubation at room temperature for two hours, cholate is removed by adding 1 g of washed Biobeads SM2 resin (BIO-RAD), with gentle agitation overnight at room temperature. myr-Arf1·GTPγS anchored on MSPΔH5 was purified by a Superdex 200 Increase 10/300 GL column (GE Healthcare) into the final D$_2$O buffer 20 mM Tris-D11 (pD 7.4) or H$_2$O buffer 20 mM Tris-HCl (pH 7.4), 150 mM NaCl, 1 mM MgCl$_2$, and 0.5 mM TCEP. The chromatography trace is shown in Supplementary Fig. 2A. The fractions containing nanodisc-anchored myr-Arf1●GTPγS were confirmed by SDS-PAGE (Supplementary Fig. 2B) and concentrated for immediate usage without freezing. The benefits of U-[$^2$H],$^{13}$CH$_3$-methyl labeled myr-Arf1 is elaborated in Supplementary Methods (Supplementary Fig. 3).

### Solution-state NMR spectroscopy

Experiments were performed using ~100 µM myr-Arf1 anchored to 50 µM nanodiscs in either H$_2$O buffer containing 20 mM Tris-HCl (pH 7.4), 1 mM MgCl$_2$,150 mM NaCl, and 0.5 mM TCEP or D$_2$O buffer where the Tris is substituted with 20 mM Tris-d$_{11}$ (pD 7.4) (Cambridge Isotopes, Inc. DLM-3593). Samples (~250 µL) were contained in 5 mm Shigemi microcells (Sigma-Aldrich, Inc. Z543349). Data were acquired at 25 °C using Bruker AVIII-850 and AVIII-800 spectrometers operating under TopSpin 3.6.5 and equipped with cryogenic TCI probes. All NMR data were processed using NMRPipe[50] and analyzed using NMRFAM-Sparky[51].

Assignments of backbone $^1$H-$^{15}$N and sidechain $^{13}$CH$_3$-ILV resonances of myr-Arf1●GTP:nanodisc were made by transfer from solution studies of L8K-Arf1[18] (BMRB 27726 (unmyristoylated L8K-Arf1, 181 residues, 5′-GTPγS)), by superposing spectra using NMRFAM-Sparky[51] and nightshift[52]. Resonances for the N-terminal helix (residues 2–23) were not observed in the $^1$H-$^{15}$N experiments; however, $^1$H-$^{13}$C resonances were observed for residues I4, L8, and L11 in the N-terminal helix. These resonances were assigned by uniqueness of only one additional

isoleucine resonance type and by mutation of L8 to identify L8 from L11 in the two additional leucine resonances compared to L8K-Arf1.

### PRE NMR measurements

Methyl-PREs are reported as the ratio of cross peak intensities from two myr-Arf1 $^1$H-$^{13}$C HMQC spectra collected from separate samples, with and without 5-doxyl PC in the nanodisc. The experiments were carried out using an echo-antiecho pulse sequence (Bruker TopSpin 3.6.5 library hmqcetgp with minor modifications for solvent suppression) with a 2 s recycle delay, 64 scans per fid, 200 complex points in the $^{13}$C dimension (20 ppm) and 2048 complex points in the $^1$H dimension (14 ppm). Typical pulse widths were 11 µs ($^1$H) and 13 µs ($^{13}$C). Thirty out of forty-four well-resolved methyl group reporters were analyzed. The error values were calculated by the formula in Eq. (1):

$$Error = \frac{I_{para}}{I_{dia}} \sqrt{\left(\frac{1}{(S/N)}\right)^2_{para} + \left(\frac{1}{(S/N)}\right)^2_{dia}} \qquad (1)$$

where $I_{para}, (S/N)_{para}$, and $I_{dia}, (S/N)_{dia}$ are the intensity and signal-noise ratios of resonance measured in paramagnetic and diamagnetic samples, respectively.

### Solid-state NMR sample preparation and acquisition

For the solid-state $^2$H NMR experiments, myr-Arf1 from *Saccharomyces cerevisiae* (Uniprot P11076) was prepared in a procedure based on the protocol according to Randazzo et al.[53] with modifications. Briefly, cells of the *E. coli* BL21(DE3) strain RSB1040 (kindly provided by Randy Schekman, Univ. California Berkeley), co-expressing Arf1 and N-myristoyl transferase 1 (NMT1, Uniprot P14743) from separate plasmids, were cultured in LB-medium to an OD$_{600}$ of 0.6 at 37 °C. 200 µM myristic acid (unlabeled, Sigma or deuterated, Eurisotop) was added to the culture medium immediately before induction. Protein expression was induced with 1 mM IPTG for 18 h at 18 °C. Cells were harvested at 4 °C by centrifugation at 5000 × g and resuspended in lysis buffer (25 mM Tris, 10 mM MgCl$_2$ pH 8.0, complemented with protease inhibitor (ROCHE cOmplete EDTA-free Protease Inhibitor Cocktail) and 10 mM β-mercaptoethanol). After cell lysis by French Press and centrifugation (20,000 × g, 4 °C), the supernatant was incubated with DEAE-Sepharose (Merck) at 4 °C for 1 h to largely remove DNA and anionic proteins. After centrifugation at 2000 × g and 4 °C, the Arf1-containing solution was concentrated using an Amicon centrifuge filter with a 10,000 kDa MWCO (Merck) and applied to a Sephacryl S100 16/60 HR size exclusion chromatography column (GE Healthcare), which was developed with 20 mM HEPES, 100 mM NaCl, 2 mM MgCl$_2$, pH 7.4. Arf1-containing fractions were again concentrated with a centrifuge filter concentrator.

For NMR spectroscopy, POPC or POPC-d$_{31}$ (both Avanti Polar Lipids) was dissolved in organic solvent, dried under vacuum (10 mbar), and dissolved in buffer (20 mM HEPES, 100 mM NaCl, 1.2 mM MgCl$_2$, 2.5 mM EDTA). After incubation at 37 °C for 30 min and 10 freeze-thaw cycles, large unilamellar vesicles were prepared by extrusion[54]. Subsequently, for the sample containing myr-Arf1, another buffer (20 mM HEPES, 100 mM NaCl, 6.25 mM EDTA, 0.45 mM GTPγS) and the d$_{27}$-myr-Arf1 solution at a 1:1.5 ratio were added to reach a 1:150 protein/lipid molar ratio. The sample was flash-frozen once to allow myr-Arf1 to bind to the liposome's inner leaflet as well. After incubation at 37 °C for 4 h, the samples were centrifuged at ~90,000 × g for 14 h. The pellets were frozen in liquid nitrogen and lyophilized under a vacuum of ~0.1 mbar. Subsequently, the samples were hydrated to 50 wt% with deuterium-depleted H$_2$O, freeze−thawed, stirred, and gently centrifuged for equilibration. The samples were then transferred to 5 mm glass vials and sealed with Parafilm for NMR measurements. $^2$H NMR spectra were acquired on a wide-bore Bruker

Avance 750 NMR spectrometer operating under TopSpin 3.5 software and at a resonance frequency of 115.1 MHz for $^2$H. A single-channel (non-cryogenic) solids probe equipped with a 5 mm solenoid coil was used. Samples were placed inside 4 mm rotors to prevent dehydration but were not rotated. All measurements were conducted at a temperature of 30 °C. The $^2$H NMR spectra were accumulated with a spectral width of ±250 kHz using quadrature phase detection, a phase-cycled quadrupolar echo sequence[55] with two ~3 µs π/2 pulses separated by a 60 µs delay, and a relaxation delay of 0.5 s. Acquisition times and number of scans for spectrum of POPC-$d_{31}$ were 4 ms and 512, respectively, while they were 2 ms and 16,384 for the spectrum of POPC-$d_{31}$ in presence of myr-Arf1 and 1.1 ms and 109,568 for the spectrum of $d_{27}$-myr-Arf1. For spectrum processing and all analysis of the spectra self-written software was used. All spectra were symmetrized. Details of the order parameter determination have been described previously[56]. A phase-cycled inversion-recovery quadrupolar echo pulse sequence was used to measure the relaxation rates for the decay of Zeeman order ($T_{1Z}$; spin-lattice relaxation time). A relaxation delay of 2 s was used, and all other parameters were the same as those for recording the $^2$H NMR spectra. Spectra at the following inversion recovery delays were acquired: 0.001, 0.007, 0.014, 0.023, 0.035, 0.07, 0.1, 0.2, 0.4, 1, 1.8 s. Acquisition times and number of scans for partially relaxed spectra of POPC-$d_{31}$ were 4 ms and 2368, respectively, while they were 1.5 ms and 3072 for the spectrum of POPC-$d_{31}$ in presence of myr-Arf1 and 2.1 ms and 12,800 for the spectrum of $d_{27}$-myr-Arf1. For determination of the relaxation rates self-written software was used, where a number of individual peaks were simultaneously fitted to all partially relaxed spectra.

### Neutron reflectometry

NR measurements were performed on the CGD-Magik reflectometer at the NIST Center for Neutron Research[57], Gaithersburg, Maryland, United States, and the Inter reflectometer at the ISIS neutron and muon source, Didcot, United Kingdom[58]. Reflectivity curves were recorded at room temperature for momentum transfer values 0.01 Å$^{-1}$ ≤ $q_z$ ≤ -0.25 Å$^{-1}$. The neutron sample cells allow in situ buffer exchange, and using this capability, series of measurements on the same bilayer under different isotopic buffers (20 mM Tris-HCl, 150 mM NaCl, 1 mM MgCl$_2$, and 5 mM DTT) in the absence and presence of proteins were performed on the same sample area. Sparsely-tethered lipid bilayer membranes (stBLMs) were prepared in the NR cell using the HC18 tether[33] by fusing vesicles of the desired lipid composition using established procedures[16,59]. After preparation of the stBLM, NR data were sequentially collected with D$_2$O and H$_2$O-based buffer. Buffer exchange was accomplished by flushing ~3 cell volumes of buffer through the sample cell using a syringe. After characterization of the as-prepared bilayer, one cell volume of Arf1 protein in H$_2$O-based buffer was introduced into the sample cell using one syringe and cell inlet. After filling the cell completely, one half cell volume of GTPγS and EDTA were introduced using a second syringe and cell inlet. Both volumes were continuously mixed by pushing the two attached syringes back and forth 100 times over 5 min. The protein concentration over both volumes was 13.3 µM. GTPγS was at 66 µM and EDTA at 1.33 mM. After 30 min, the cell was rinsed with pure buffer that contained 5 mM MgCl$_2$ instead of 1 mM. After 5 min, the cell was rinsed with standard buffer and subsequent NR measurements in D$_2$O and H$_2$O-based buffer were performed.

NR datasets collected on stBLMs immersed in isotopically different buffers (Supplementary Figs. 4–6), before and after protein addition, were analyzed simultaneously (4 datasets per stBLM). One-dimensional structural profiles of the substrate and the lipid bilayer along the interface normal z were parameterized with a model that utilizes continuous volume occupancy distributions of the molecular components[60]. For NR datasets measured after protein addition, free-form Arf1 profiles were modeled using Hermite splines with control points on average 15 Å apart[28]. The extension of the protein along the membrane normal determines the number of spline control points and was iteratively refined. A Monte Carlo Markov Chain-based global optimizer was used to determine best-fit parameters and their confidence limits[59].

### Molecular dynamics simulation

The simulation system was constructed using the residue range 2–181 of the human Arf1 construct (Uniprot P84077). The protein was modeled using a close analog from yeast (PDB: 2KSQ) which is 76% identical[3]. All simulations were performed in the GTP-bound conformation with a myristoyl tail at the N-terminus. All hydrogen atoms and the terminal patches were added using the PSFGEN plugin of VMD (Visual Molecular Dynamics)[61].

To capture membrane association, we performed membrane-binding simulations by employing a highly mobile membrane mimetic (HMMM) model. Multiple independent HMMM simulation systems were constructed using HMMM BUILDER in CHARMM-GUI[62], ensuring different initial arrangements of lipids. Using short-tailed lipids and the organic liquid DCLE that mimics the bilayer interior, HMMM models enhance lipid diffusion and lipid reorganization in the membrane, thereby allowing faster spontaneous binding of peripheral proteins[35]. This approach has been extensively used to study a variety of peripheral and integral membrane proteins[63–67]. With the aid of the HMMM membrane model, we were able to perform multiple membrane-binding simulations of GTP-bound myristoylated human Arf1 in the presence of mixed-lipid membranes containing phosphatidylcholine (POPC), phosphatidylserine (POPS), and PI(4,5)P$_2$ in varying compositions (see Supplementary Table 2).

At the start of the membrane-binding simulations, the N-terminal helix of Arf1 was placed above the surface of the membrane, with the myristoyl tail, which is known to insert into the membrane, placed below the plane of the phosphorous atoms of the cis leaflet. Each simulation started with randomized initial velocities, and each replica was simulated for 100 ns. In addition, to sample the conformational dynamics of the G domain on the membrane surface, each membrane-bound replica was converted to a full-membrane system and simulated for additional 1000 ns.

All the simulations were performed under periodic boundary conditions using NAMD2[68,69], CHARMM36m protein and lipid forcefields[70,71] and TIP3P water[72]. Short trailed in HMMM simulations are best simulated in fixed area ensemble. Thus, all the membrane-binding HMMM simulations were performed in NP$_n$AT ensembles at 1 atm and 37 °C. Non-bonded interactions were calculated with 12 Å cutoff and a switching distance of 10 Å. All the long-range electrostatic interactions were calculated using the particle mesh Ewald (PME) method[73]. A constant temperature was maintained by Langevin dynamics with a damping coefficient of 0.5 ps$^{-1}$ applied to all the atoms, and constant pressure was maintained using the Nosé–Hoover Langevin piston method[74]. An integration step of 2 fs was used in all the simulations.

### PRE and MD data analysis

Descriptions of data analysis procedures for PRE and MD are provided in the Supplementary Information.

### Statistical analysis

Data are expressed as means ±σ or ±SD when applicable. Representative data from two or more independent experiments were analyzed. P-values were determined using the t-test.

### Reporting summary

Further information on research design is available in the Nature Portfolio Reporting Summary linked to this article.

## Data availability

The datasets generated and analyzed in the current study and presented as the main and supplementary figures are available in the Figshare repository with the identifier https://doi.org/10.6084/m9.figshare.23846991. All other data that support the findings of this study will be available from the corresponding authors upon reasonable request. Two PDB depositions are referenced in this study and are available from the PDB repository as follows: PDB: 3LVQ (ASAP3-Arf6 fusion construct comprised of residues 9–175 of Arf6 (Uniprot P62330) with residues 416–702 of ASAP3 (Uniprot Q8TDY4)); PDB: 2KSQ (Arf1 residues 2–181 (Uniprot P11076)). One BMRB deposition is referenced and is available as follows: BMRB 27726 (unmyristoylated L8K-Arf1, 181 residues, 5′-GTPγS)), All other data that support the findings of this study will be available from the corresponding authors upon reasonable request.

## Code availability

The HMMM software and molecular dynamics programs utilizing NAMD2 and CHARMM36 force field has been described previously[35] and inquiries should be addressed to E.T. The custom software developed to analyze PRE data is available at Zenodo [https://doi.org/10.5281/zenodo.10034880].

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

## Acknowledgements

The authors acknowledge the use of the Biophysics Resource, Center for Structural Biology, NCI, and the assistance of Dr. Sergey Tarasov and Ms. Marzena Dyba. The research was supported by the Intramural Research Program of the National Cancer Institute (Projects ZIA BC 011419, ZIA BC 011131, and ZIA BC 011132 supported O.S., Y.Z., J.L., and R.A.B.; Project BC007365 supported P.A.R.). The computational component of this study was supported by the NIH under Awards P41-GM104601 (to E.T.) and R01-GM123455 (to E.T.). We also acknowledge computing resources provided by Blue Waters at the National Center for Supercomputing Applications and Extreme Science and Engineering Discovery Environment (Award MCA06N060 to E.T.). S.P. would like to thank the Beckman Institute Graduate Fellowship for funding. The content is solely the responsibility of the authors and does not necessarily represent the official views of the NIH. F.H. and M.L. were supported by the U.S. Department of Commerce, Award 70NANB17H299. The research was performed, in part, at the National Institute of Standards and Technology (NIST) Center for Nanoscale Science and Technology. Certain commercial materials, equipment, and instruments are identified in this work to describe the experimental procedure as completely as possible. In no case does such an identification imply a recommendation or endorsement by NIST, nor does it imply that the materials, equipment, or instrument identified is necessarily the best available for the purpose.

## Author contributions

Y.Z., O.S., and S.P. contributed equally to this work. Y.Z. and O.S. conducted all solution-state NMR experiments and analyses. F.H. and L.A.C. conducted all NR experiments and analyses, performed. A.V, S.D., K.B., and D.H. conducted all solid-state NMR experiments and analyses, S.P. conducted all MD simulations and analyses. J.L. and Y.L. contributed to the design and engineering of all protein systems. P.A.R., M.L., E.T., and R.A.B. conceived and designed the project and provided funding and resources. Y.Z., O.S., S.P., F.H. A.V., D.H., P.A.R., E.T., and R.A.B. wrote and edited the final manuscript.

## Funding

## Competing interests

The authors declare no competing interests during the conduct of this research. Subsequent to this research, S.P. has become an employee of Loxo @ Lilly and is a shareholder of stock in Eli Lilly and Co.; however, those activities are separate and distinct from this research report.
