## [Peer Review File · Nature Communications]

Myr-Arf1 conformational flexibility at the membrane surface sheds light on the interactions with ArfGAP ASAP1REVIEWER COMMENTS

Reviewer #1 (Remarks to the Author):

In this study, Zhang, Soubias, Pant et al combine a range of biophysical methods (NMR, NR, MD) to study the membrane bound state of Arf1 with the membrane. This area is typically difficult to study using structural biology, so the combination of biophysical approaches as used here is appropriate. The study is well carried out and the data has clearly been rigorously gathered and analyzed. The correct tools and analyses have been applied and the findings are in general justified from the data.

The manuscript is very technical, and it's not clear to me how much biological insight there is. It is, in places, also hard to follow, and I'm not sure how much it will appeal to the broad audience of Nat Comms.

Some specific comments:

- Seeing as the study is in essence doing structural biology, coordinates from different states seen in the MD should be made available for readers to access.
- The style varies dramatically from figure to figure (e.g. font style, font size, color scheme). This doesn't impact the quality of the data but affects the flow of the manuscript considerably
- Why does the distribution of the tether change so much between 2B and C? I can't see this discussed – in fact, I don't see much mention of the data in panel C at all.
- The coloring between 3A and 3B seems incorrect – PS is green in A but the PS membrane is blue in B?
- Figure 3 – what is the meaning of 'cis phosphorus plane'?
- How has convergence been demonstrated for the 2D PMFs in Figure 4? For instance, how do the landscapes change if only a subset of simulations are analyzed?
- In the MD, do individual repeats sample more than 1 state? If so, how much time do they spend in the state, i.e. what are the kinetics of the changes? If not, how can we be sure that there are enough replicates to be converged?
- The boxes for S1 and S3 in 4A need to not be the same color as the landscape. S3 in particular is very hard to see.
- There are a few visual glitches on the TOC graphic
- There are a few instances where the word "REF" or "ref" have been left in
- Supplemental Figure 8 has some Y axes label issues

Reviewer #2 (Remarks to the Author):

This manuscript presents a very detailed description of the multiple conformations adopted by myristoylated ADP-ribosylation factor-1 (mry-Arf-1) anchored to a membrane nanodisc, and the probable selection of one of these conformers when bound to the PH domain of an Arf GTPase activating protein.

This is an important regulatory process of potentially broad interest to a biochemistry/cell biology audience. The research is based on a modern combination of structural biology tools, including NMR, Neutron Reflectometry, and Molecular Dynamics modeling; all well done and well-presented.

I have only a few relatively minor comments/suggestions.

- 1) On line 71 of page 4 the authors have left an unused reference marker [ref] in the manuscript. Referencing is quite complete, but it may be appropriate to include a 2014 paper by Liu et al. that characterizes Arf interactions with another PH domain, that from the adaptor protein, Fapp1.
- 2) On Page 12 – lines 247. Deviation between populations observed by a PRE-based ensemble selection and conformer populations in MD simulations are not surprising. MD force fields are not sufficiently accurate do this accurately. The authors may want to specifically point to MD limitations.
- 3) On page 13. The authors may want to add some description of Pake doublet features. These are not frequently presented to a general audience.
- 4) On line 286 of page 13 – something may be missing; “calculate chain extension profiles that the distance of individual carbon positions”
- 5) At the end of the first paragraph on page 14, the summary statement about chain extension seems to contradict the prior discussion emphasizing mobility. This might be re-worded.
- 6) The model presented at the bottom of Figure 7 should probably have its own designating letter. It is unfortunate that resolution prevented identification of Arf interaction surfaces. However, given the authors previous work identifying interaction surfaces of the PH domain, it would seem logical to include these on a ribbon diagram instead of the ellipse currently in the figure. If there are discrepancies with the current model, these should be discussed.

Reviewer #3 (Remarks to the Author):

The manuscript by Zhang et al. describes the conformation of the protein Myr-Arf1 in its membrane bound state using a combination of nuclear magnetic resonance (NMR), neutron reflectometry (NR) and molecular dynamics (MD) simulations. The presented study extends previous simulation, NMR and NR work that used only the myristoylated N-terminal helix. The manuscript is well-written and describes clearly the complementary information obtained using two different experimental techniques as well as analysis of the simulations. Overall, the study gives a detailed account of the work, and is a good example of the challenges involved in combining information from experimental techniques that have intrinsically a very different resolution in time and space than computer simulations. The primary structural finding that the protein interacts with the membrane lipids through the myristoyl chain and its amphipathic helix is however quite a general mechanism for peripheral protein association to membranes and is as such not surprising. The main result presented is that the membrane bound protein explores a number of configurations that expose different surfaces for binding interactions with proteins activating Arf, but it appears that this was identified mainly from the MD simulations, rather than the combination of techniques. It is unfortunate that the binding of ASAP1-PH with the membrane-bound myr-Arf-GTP was not also investigated using NR to observe the mode of binding in addition to the effect on the rotational dynamics, which would have strengthened the conclusion. Conceptually the work and analysis is sound, and the results support the conclusions fairly. The biophysical consequences

of the three states are discussed in some detail but without reference to their significance to function, or the proposed relevance in cancer. My impression is that the work in the manuscript, while being of very good quality, does not present a significant advance for the field, or that the current discussion in the manuscript is lacking the type of broader context that is needed to demonstrate this.

Specific comments:

Line 71 – please specify the missing reference denoted only as [ref].

Line 81-82: replace “N-term” with “N-terminally”

That the protein associates with membranes through the myristoyl chain and the amphi

Line 401-402: the statement that “The NMR, MD, and NR data collectively reveal that the G domain conformational space populates states (S1, S2, and S3) that can present interfaces...” is somewhat misleading as it seems that the three different states were really only identified in the simulations, and that mainly one can say that the NR/NMR data are not inconsistent with them.

Reviewer #4 (Remarks to the Author):

This well-written manuscript describes combined results from NMR, MD and NR that shed light on the conformation and dynamics of membrane-associated myristoylated Arf1 (myr-Arf1) and its interactions with the PH domain of ASAP1.

The study is rigorously executed, and provides the first view of myr-Arf1 in a phospholipid bilayer membrane free of the deleterious effects of detergents. Contrary to previous studies that relied on detergent-lipid mixed micelles or bicelles, the present results from nanodiscs show that the myristoyl chain of Arf1 inserts in the lipid bilayer, and contributes to the overall dynamics of the globular protein domain. The nanodisc platform also enables the authors to probe the effects of charged lipids which are appreciable and important for regulating protein dynamics and orientation at the membrane surface, predisposing it for ligand recognition.

The combination of approaches is appropriate and interesting, and the correspondence between experimental and MD simulation data is very compelling.

The work is noteworthy and significant because it provides initial key insights about the way in which myr-Arf1 dynamics regulates ligand binding at the membrane surface, and sets the stage for investigation of complex assemblies of membrane-associated Arf1 and its ligands.

I have only minor comments.

Fig. 2 – Suggest revision to align panels C-E with the bars in panel B. This would make it easier to follow.

Fig. 7 – I couldn't tell if the NMR spectra have been assigned. If yes, is it possible to map the peak intensity (+/- ASAP1-PH domain) to gain additional insights about the binding and recognition interaction?

p.4 line 71 – Delete the term "[ref]".

p.9 line 175 – Delete duplicate "we employed".

Response to Reviewers

Reviewer #1 (Remarks to the Author):

In this study, Zhang, Soubias, Pant et al combine a range of biophysical methods (NMR, NR, MD) to study the membrane bound state of Arf1 with the membrane. This area is typically difficult to study using structural biology, so the combination of biophysical approaches as used here is appropriate. The study is well carried out and the data has clearly been rigorously gathered and analyzed. The correct tools and analyses have been applied and the findings are in general justified from the data.

Response: We thank reviewer for the positive assessment of the study.

The manuscript is very technical, and it's not clear to me how much biological insight there is. It is, in places, also hard to follow, and I'm not sure how much it will appeal to the broad audience of Nat Comms.

Response: We acknowledge the reviewers concerns, and we respectfully contend that the subject of structural insights for peripheral membrane proteins is both broadly significant and extremely challenging. The integrated approach of experimental and computational methods combines to establish the baseline understanding that will evolve in subsequent ongoing studies of the small GTPases and GAP proteins complexed at the membrane surface. We are continuing work with our integrated approach as stated below in our response to Reviewer #3. It may be envisioned that characterization of the multiple states could be leveraged to design/discover small molecule interactors, which could stabilize Arf1 into an inactive state, thus disallowing effector binding.

Some specific comments:

- Seeing as the study is in essence doing structural biology, coordinates from different states seen in the MD should be made available for readers to access.

Response: Molecular dynamics snapshots taken from three independent trajectories of Arf1 on the surface of lipid bilayer containing PC-PS-PIP2 lipids have been deposited in the figshare project associated with this submission (link: [10.6084/m9.figshare.23846991](https://www.figshare.com/projects/10.6084/m9.figshare.23846991)). This will be made public upon acceptance.

- The style varies dramatically from figure to figure (e.g. font style, font size, color scheme). This doesn't impact the quality of the data but affects the flow of the manuscript considerably

Response: We appreciate the reviewers comments, and we have reviewed and updated the figures to have a consistent style.

• Why does the distribution of the tether change so much between 2B and C? I can't see this discussed – in fact, I don't see much mention of the data in panel C at all.

Response: Data for panel 2C was collected at the ISIS neutron source at the Rutherford Appleton Laboratory in the UK. The other data were collected at the NIST Center for Neutron Research in the USA. The surface roughness of the scattering substrates provided by both facilities differs by about 2 Å RMS, leading to broader volume occupancy distributions for the ISIS sample, which is the main visual difference between Figure 2C and the others. Table S1 was updated to provide roughness parameters. RMS does not significantly affect peak positions and the amount of material discussed in the text. Variations in the tether densities can additionally contribute to the visual differences.

• The coloring between 3A and 3B seems incorrect – PS is green in A but the PS membrane is blue in B?

Response: We thank the reviewer for pointing out this inconsistency in Figure 3. We have accordingly updated Figure 3 (panel A) in the revised manuscript (also shown below).

Figure 3: Dynamics of the G domain in membrane-bound Arf1. (A) Top view of a representative Arf1 conformation in a lipid bilayer containing PC, PS, and PI(4,5)P₂ lipids, highlighted in different colors. (B) Average COM distance of the G domain (residues 17-181) from the lipid-bilayer (phosphorous plane set at 0), captured from the HMMM (light bars) and full-membrane simulations (dark bars) in PC-PS-PI(4,5)P₂, PC-PI(4,5)P₂, and PC

membranes. (C-E) Representative C_{α} membrane insertion profile of Arf1, averaged over the last 50 ns of the nine HMMM membrane-binding replicas for (C) pure PC, (D) PC-PI(4,5)P₂, and (E) PC- PS-PI(4,5)P₂ membranes. The dashed line in each graph represents the position of the cis phosphorus plane (phosphorous atoms of the upper leaflet on which Arf1 was initially placed).”

- Figure 3 – what is the meaning of ‘cis phosphorus plane’?

Response: Cis phosphorous plane corresponds to the phosphorous atoms in the leaflet closer to Arf1 (side of the membrane where Arf1 was placed). We have updated the caption of Figure 3 in the revised manuscript:

“Figure 3: Dynamics of the G domain in membrane-bound Arf1. (A) Top view of a representative Arf1 conformation in a lipid bilayer containing PC, PS, and PI(4,5)P₂ lipids, highlighted in different colors. (B) Average COM distance of the G domain (residues 17-181) from the lipid-bilayer (phosphorous plane set at 0), captured from the HMMM (light bars) and full-membrane simulations (dark bars) in PC-PS-PI(4,5)P₂, PC-PI(4,5)P₂, and PC membranes. (C-E) Representative C_{α} membrane insertion profile of Arf1, averaged over the last 50 ns of the nine HMMM membrane-binding replicas for (C) pure PC, (D) PC-PI(4,5)P₂, and (E) PC- PS-PI(4,5)P₂ membranes. The dashed line in each graph represents the position of the cis phosphorus plane (phosphorous atoms of the leaflet closer to where Arf1 was initially placed).”

- How has convergence been demonstrated for the 2D PMFs in Figure 4? For instance, how do the landscapes change if only a subset of simulations are analyzed?

Response: As suggested, in the revised manuscript, we have added 2D PMF plots (Fig S10) calculated by considering only a subset of simulation data. The figure (see below) shows that the data from 50%, 75%, or 90% of the trajectories result in almost identical maps, thus indicating convergence.

Supplemental Figure 11. Convergence of 2D PMF in three different lipid environment was assessed by recalculating them considering only a subset of simulation data. 2D PMF plots corresponding to PC-PS-PIP2 containing lipid bilayer by considering (A) 50%, (B) 75% and (C) 90% of the simulation data. 2D PMF plots corresponding to PC-PIP2 containing lipid bilayer by considering (D) 50%, (E) 75%, and (F) 90% of the simulation data. 2D PMF plots corresponding to pure PC containing lipid bilayer (G) 50%, (H) 75%, and (I) 90% of the simulation data.

• In the MD, do individual repeats sample more than 1 state? If so, how much time do they spend in the state, i.e. what are the kinetics of the changes? If not, how can we be sure that there are enough replicates to be converged?

Response: In the presence of PC-PC-PIP2 containing lipid bilayer, 3 trajectories exhibit the indication that more than 1 state might be sampled. However, extracting reliable kinetics from the simulations will require orders of magnitude longer simulations (~100 microseconds, as done in previous studies, e.g., in ref. 42 of Supplementary Information). This can be certainly an interesting topic for future studies. Similarly in the presence of PC-PIP2 containing lipid bilayer, we observed 3 trajectories where more than 1 state were sampled.

- The boxes for S1 and S3 in 4A need to not be the same color as the landscape. S3 in particular is very hard to see.

Response: In Figure 4, we have adjusted the labeling to improve clarity of the minima in panels 4A and 4E.

- There are a few visual glitches on the TOC graphic

Response: We appreciate the reviewer pointing out this earlier draft figure. It is replaced in the revision with a corrected, high-resolution image.

- There are a few instances where the word “REF” or “ref” have been left in

Response: We apologize for these being left in the manuscript file and have removed them.

- Supplemental Figure 8 has some Y axes label issues

Response: We thank reviewer for pointing out this issue. Figure S8 has been updated in the revised manuscript:

Supplemental Figure 8. Ensemble-averages $C\alpha$ profiles of Arf1 residue insertion into the membrane calculated during the last 50 ns of the 9 replicas of the HMMM membrane-binding simulations in the presence of POPC:POPS:PI(4,5)P2 (8:1.5:0.5) lipid bilayers. In all

the simulations, the N-terminal helix was observed to penetrate into the membrane below the phosphorus plane (dashed lines) of the lipid bilayer. Standard deviations are shown as blue bars.

Reviewer #2 (Remarks to the Author):

This manuscript presents a very detailed description of the multiple conformations adopted by myristoylated ADP-ribosylation factor-1 (mry-Arf-1) anchored to a membrane nanodisc, and the probable selection of one of these conformers when bound to the PH domain of an Arf GTPase activating protein. This is an important regulatory process of potentially broad interest to a biochemistry/cell biology audience. The research is based on a modern combination of structural biology tools, including NMR, Neutron Reflectometry, and Molecular Dynamics modeling; all well done and well-presented.

Response: We would like to thank reviewer for the positive assessment of our study.

I have only a few relatively minor comments/suggestions.

1) On line 71 of page 4 the authors have left an unused reference marker [ref] in the manuscript. Referencing is quite complete, but it may be appropriate to include a 2014 paper by Liu et al. that characterizes Arf interactions with another PH domain, that from the adaptor protein, Fapp1.

Response: We apologize for this notation being left in the manuscript file, and it has been removed. We appreciate the suggestion regarding Liu et al., and this citation has been added in line 98.

2) On Page 12 – lines 247. Deviation between populations observed by a PRE-based ensemble selection and conformer populations in MD simulations are not surprising. MD force fields are not sufficiently accurate do this accurately. The authors may want to specifically point to MD limitations.

Response: We have added a comment highlighting that the differences in the conformer populations observed in MD and PRE-based measurement might be attributed to the MD force fields:

“However, the MD simulations suggest a larger population of conformers in the S1 state than the PRE ensemble analysis selects to fit the experimental data. The observed discrepancies between the MD simulations and PRE-based ensemble selection might arise due to force field limitations in accurately capturing non-covalent interactions.”

3) On page 13. The authors may want to add some description of Pake doublet features. These are not frequently presented to a general audience.

Response: We added a sentence explaining that line shape on page 13, lines 276-280.

In such spectra, Pake-doublets are observed for each labeled position of the hydrocarbon chain. These Pake-doublets are symmetric line shapes that represent all possible orientations of a C-D bond vector in a powder-type sample and the distance between the two most intense peaks of the powder pattern can directly be related to the segmental order parameter, which describes the amplitude of the motion of the corresponding C-H bond.

4) On line 286 of page 13 – something may be missing; “calculate chain extension profiles that the distance of individual carbon positions”

Response: Indeed, we elaborated and corrected this statement, see page 14, line 297.

5) At the end of the first paragraph on page 14, the summary statement about chain extension seems to contradict the prior discussion emphasizing mobility. This might be re-worded.

Response: We have rewritten this statement, see page 14, lines 301-306.

Therefore, to reliably compare the chain extension profiles for two different molecules, its slope should be considered. In this case the slope of the chain extension profiles of the myr-Arf1 myristoyl chain is visibly smaller than that of the palmitoyl chain of POPC. This indicates that the myristoyl chain of Arf1 is immersed in the bilayer and is somewhat compressed along the membrane normal and therefore exhibits greater lateral mobility compared to the palmitoyl chain of POPC.

6) The model presented at the bottom of Figure 7 should probably have its own designating letter. It is unfortunate that resolution prevented identification of Arf interaction surfaces. However, given the authors previous work identifying interaction surfaces of the PH domain, it would seem logical to include these on a ribbon diagram instead of the ellipse currently in the figure. If there are discrepancies with the current model, these should be discussed.

Response: We appreciate the reviewer’s comments. We intend for panel C to represent the dynamic nature of the G-domain represented by the three states in Blue, Orange and Green which may form a complex with the ASAP1-PH domain (shown in the lower part of panel C), wherein the switch 1 and 2 are highlighted in yellow in both the upper and lower parts of panel C. With respect to illustrating the binding surface on ASAP1-PH, our previous work (Soubias et al., *Sci. Adv.* 2020) showed interaction of soluble ^{L8K}Arf1•GTP with ASAP1-PH. Ongoing work is establishing the interface at the membrane surface and will be reported separately.

Reviewer #3 (Remarks to the Author):

The manuscript by Zhang et al. describes the conformation of the protein Myr-Arf1 in its membrane bound state using a combination of nuclear magnetic resonance (NMR), neutron reflectometry (NR) and molecular dynamics (MD) simulations. The presented study extends previous simulation, NMR and NR work that used only the myristoylated N-terminal helix. The manuscript is well-written and describes clearly the complementary information obtained using two different experimental techniques as well as analysis of the simulations. Overall, the study gives a detailed account of the work, and is a good example of the challenges involved in combining information from experimental techniques that have intrinsically a very different resolution in time and space than computer simulations. The primary structural finding that the protein interacts with the membrane lipids through the myristoyl chain and its amphipathic helix is however quite a general mechanism for peripheral protein association to membranes and is as such not surprising. The main result presented is that the membrane bound protein explores a number of configurations that expose different surfaces for binding interactions with proteins activating Arf, but it appears that this was identified mainly from the MD simulations, rather than the combination of techniques. It is unfortunate that the binding of ASAP1-PH with the membrane-bound myr-Arf-GTP was not also investigated using NR to observe the mode of binding in addition to the effect on the rotational dynamics, which would have strengthened the conclusion. Conceptually the work and analysis is sound, and the results support the conclusions fairly. The biophysical consequences of the three states are discussed in some detail but without reference to their significance to function, or the proposed relevance in cancer. My impression is that the work in the manuscript, while being of very good quality, does not present a significant advance for the field, or that the current discussion in the manuscript is lacking the type of broader context that is needed to demonstrate this.

Response: We value the reviewer's assessment. As stated in response to Reviewer 1, the ability to structurally characterize peripheral membrane proteins and their complexes is exceedingly challenging. The dynamics that we characterize in this study interferes with and precludes investigation by X-ray crystallography and cryoEM. The combined analyses, using the computational and experimental methods of MD, NMR, and NR, provides important understanding of the initial state of myr-Arf1•GTP associated with the membrane surface. Attempts to examine the myr-Arf1•GTP:ASAP1-PH system with NR have not yet succeeded. However, the NMR data provide a clear indication that this state changes significantly upon complexation with the ASAP1-PH domain. The changes observed suggest effects in both the G-domain and the ASAP1-PH domain (from our earlier reports). The complex is an increasingly difficult system to evaluate for interfaces and functional consequences; however, our ongoing efforts to provide detailed insights using all methods in our integrated approach are progressing, and we feel this is currently beyond the scope of the manuscript.

Specific comments:

Line 71 – please specify the missing reference denoted only as [ref].

Response: We apologize for this notation being left in the manuscript file, and it has been removed.

Line 81-82: replace “N-term” with “N-terminally”

Response: We have made this correction.

That the protein associates with membranes through the myristoyl chain and the amphi

Response: This text was ‘floating’ in the reviewer comment portion of the email report. It is not clear if this was an editing error, hence we have left this without comment.

Line 401-402: the statement that “The NMR, MD, and NR data collectively reveal that the G domain conformational space populates states (S1, S2, and S3) that can present interfaces...” is somewhat misleading as it seems that the three different states were really only identified in the simulations, and that mainly one can say that the NR/NMR data are not inconsistent with them.

Response: We respectfully contend that both the NR and NMR data reveal a dynamic interconversion of states, which correlates well with the MD data. The NR data illustrate that there is not a single state but a dynamic average (cf. page 8, lines 158-160; response to Reviewer #1). The NMR PRE data are analyzed to reveal an ensemble populated by the three states seen in the MD results. The analyses reveal that equal population of all possible conformations in the MD trajectories do not fit the observed NMR data (Figure S13); however, the ensemble analyses show that sets of conformations selected from the set of all possible conformations do provide accurate fits of the data. These unbiased, selected sets correspond to those observed in the MD PMF plots (figure 4), as shown in Figure 5.

Edits were made to the text in lines 232-236, page 11 to refer to these points.

Reviewer #4 (Remarks to the Author):

This well-written manuscript describes combined results from NMR, MD and NR that shed light on the conformation and dynamics of membrane-associated myristoylated Arf1 (myr-Arf1) and its interactions with the PH domain of ASAP1.

The study is rigorously executed, and provides the first view of myr-Arf1 in a phospholipid bilayer membrane free of the deleterious effects of detergents. Contrary to previous studies that relied on detergent-lipid mixed micelles or bicelles, the present results from nanodiscs show that the myristoyl chain of Arf1 inserts in the lipid bilayer, and contributes to the overall dynamics of the globular protein domain. The nanodisc platform also enables the authors to probe the effects of charged lipids which are appreciable and important for regulating protein dynamics and orientation at the membrane surface, predisposing it for ligand recognition.

The combination of approaches is appropriate and interesting, and the correspondence between experimental and MD simulation data is very compelling.

The work is noteworthy and significant because it provides initial key insights about the way in which myr-Arf1 dynamics regulates ligand binding at the membrane surface, and sets the stage for investigation of complex assemblies of membrane-associated Arf1 and its ligands.

Response: We thank the reviewer for the positive assessment of the manuscript.

I have only minor comments.

Fig. 2 – Suggest revision to align panels C-E with the bars in panel B. This would make it easier to follow.

Response: We believe the reviewer is referring to Figure 3. We have made updates to figure 3, in the context to making clarifications in response to Reviewer 1. The layout of the figure may defer to publication editors for optimal formatting.

Fig. 7 – I couldn't tell if the NMR spectra have been assigned. If yes, is it possible to map the peak intensity (+/- ASAP1-PH domain) to gain additional insights about the binding and recognition interaction?

Response: The ^1H - ^{15}N and ^1H - ^{13}C -methyl-ILV NMR spectra of the Arf1 G-domain have been assigned in ref 17, cited on page 5, line 105 and page 6, line 111. The ^1H - ^{13}C Methyl-TROSY spectra assignments are provided in Figure 1C, and we have improved this figure to make the assignments more readable. The ^1H - ^{15}N and ^1H - ^{13}C assignments are made by transfer from the assignments of L8K-Arf1 that are deposited in BMRB entry 27726. The citation of the BMRB entry was added to the SI, page 12. As seen in Figure 7A, unfortunately, most of the ^1H - ^{15}N signals disappear upon binding. We are exploring methods to directly report on this interface, and this will be the subject of a subsequent manuscript.

p.4 line 71 – Delete the term "[ref]".

Response: We apologize for this notation being left in the manuscript file, and it has been removed.

p.9 line 175 – Delete duplicate "we employed".

Response: Thank you for the careful reading. We have edited this text as suggested.

REVIEWERS' COMMENTS

Reviewer #1 (Remarks to the Author):

The authors have satisfactorily addressed my comments, including on the biological importance of the study. I now fully support publication.

Reviewer #3 (Remarks to the Author):

Having read the revised manuscript and response of the authors to the comments, I cannot see any attempt to address my main concern about the lack of a broader context for the work presented. I do not at all disagree with the authors on the importance or difficulty in doing such detailed characterisation, however, as it is presented the manuscript is better suited to a specialist biophysical journal, and my main comment remains the same: "Conceptually the work and analysis is sound, and the results support the conclusions fairly. The biophysical consequences of the three states are discussed in some detail but without reference to their significance to function, or the proposed relevance in cancer. My impression is that the work in the manuscript, while being of very good quality, does not present a significant advance for the field, or that the current discussion in the manuscript is lacking the type of broader context that is needed to demonstrate this."

Response to reviewers:

Reviewer #1 (Remarks to the Author):

The authors have satisfactorily addressed my comments, including on the biological importance of the study. I now fully support publication.

We appreciate the positive response and support of our work.

Reviewer #3 (Remarks to the Author):

Having read the revised manuscript and response of the authors to the comments, I cannot see any attempt to address my main concern about the lack of a broader context for the work presented. I do not at all disagree with the authors on the importance or difficulty in doing such detailed characterisation, however, as it is presented the manuscript is better suited to a specialist biophysical journal, and my main comment remains the same: "Conceptually the work and analysis is sound, and the results support the conclusions fairly. The biophysical consequences of the three states are discussed in some detail but without reference to their significance to function, or the proposed relevance in cancer. My impression is that the work in the manuscript, while being of very good quality, does not present a significant advance for the field, or that the current discussion in the manuscript is lacking the type of broader context that is needed to demonstrate this."

We understand part of the reviewer's viewpoint in wanting to have a clear and complete outcome. However, we also appreciate that the reviewer recognizes the challenges in understanding such dynamic complexes of peripheral membrane proteins at the membrane surface. Indeed, the fact that the system is not amenable to traditional methods and requires the type of integrated approach, we contend, establishes some of the breadth of interest and impact. The approach itself is significant and illustrates the interplay between experimental and computational methods. A goal, looking forward, is that the indication of some set of discreet conformers might provide unique opportunities to assist in rational design of small molecule modulators. Furthermore, monitoring the shift in populations or selection of conformations as the system moves forward on the signaling pathway requires this type of study and elucidation as foundational knowledge.